# Genome-scale analysis of interactions between genetic perturbations and natural variation

Joseph J. Hale [1], Takeshi Matsui [2,7], Ilan Goldstein [1,7], Martin N. Mullis [1], Kevin R. Roy [3,4], Christopher Ne Ville[1], Darach Miller[2], Charley Wang[1], Trevor Reynolds[1], Lars M. Steinmetz [3,4,5], Sasha F. Levy [2,6] ✉ & Ian M. Ehrenreich [1] ✉

Interactions between genetic perturbations and segregating loci can cause perturbations to show different phenotypic effects across genetically distinct individuals. To study these interactions on a genome scale in many individuals, we used combinatorial DNA barcode sequencing to measure the fitness effects of 8046 CRISPRi perturbations targeting 1721 distinct genes in 169 yeast cross progeny (or segregants). We identified 460 genes whose perturbation has different effects across segregants. Several factors caused perturbations to show variable effects, including baseline segregant fitness, the mean effect of a perturbation across segregants, and interacting loci. We mapped 234 interacting loci and found four hub loci that interact with many different perturbations. Perturbations that interact with a given hub exhibit similar epistatic relationships with the hub and show enrichment for cellular processes that may mediate these interactions. These results suggest that an individual's response to perturbations is shaped by a network of perturbation-locus interactions that cannot be measured by approaches that examine perturbations or natural variation alone.

Genetic perturbations can interact with genetic variants (or loci) segregating in a population[1]. These interactions cause perturbations to show different phenotypic effects in genetically distinct individuals (also known as genetic background effects)[2]. For example, disease mutations often exhibit incomplete penetrance and variable expressivity in humans[3]. Similarly, background effects could cause genome editing to have variable efficacy, as applied to human disease, productivity of crops or livestock, and engineering of cells and microorganisms for industrial applications. In an evolutionary context, the fitness effect of de novo mutations can depend on interactions with specific alleles present in an individual. These interactions can impact

the dynamics of beneficial mutations spreading through a population[4,5] or deleterious mutations persisting within a population[6].

Interactions between genetic perturbations and loci also impact the phenotypic effects of loci. A perturbation may alter the degree to which an interacting locus influences a trait (magnitude epistasis) or which allele of that locus confers a higher trait value (sign epistasis)[7,8]. In some cases, a perturbation can uncover cryptic loci that do not typically show phenotypic effects[9,10], or mask loci that usually exhibit phenotypic effects[11]. The net effect of a perturbation can depend on epistasis with multiple loci that interact not only with the perturbation but also each other (higher-order epistasis)[12]. That is, the response of a

[1]Department of Biological Sciences, Molecular and Computational Biology Section, University of Southern California, Los Angeles, CA 90089, USA. [2]SLAC National Accelerator Laboratory, Menlo Park, CA 94025, USA. [3]Stanford Genome Technology Center, Stanford University, Palo Alto, CA, USA. [4]Department of Genetics, Stanford University School of Medicine, Stanford, CA, USA. [5]European Molecular Biology Laboratory, Genome Biology Unit, Heidelberg, Germany. [6]BacStitch DNA, Los Altos, CA, USA. [7]These authors contributed equally: Takeshi Matsui, Ilan Goldstein. ✉e-mail: sasha@bacstitchdna.com; ian.ehrenreich@usc.edu

given individual to a perturbation will depend on its genotype at all interacting loci[13].

While numerous examples of background effects have been reported[14–29], we lack a detailed understanding of interactions between genetic perturbations and genetic backgrounds. For example, we do not yet know what characteristics of perturbations and individuals' genotypes increase the likelihood of interactions, what types of epistasis are most likely to underlie background effects, and the properties of interaction networks between perturbations and loci. To answer these questions systematically, we require two things: perturbations at a genomic scale and extremely high-throughput measurement of the phenotypic effects of these perturbations across many individuals. Studies of this scale have been technically challenging and cost prohibitive.

Here, we developed a scalable platform for assaying interactions between genetic backgrounds and genetic perturbations using a genomic double barcoding system[30,31] and inducible CRISPR interference (CRISPRi)[32–34]. We utilized 169 haploid progeny (segregants) from a cross between the BY4716 (BY) lab and 322134S (3S) clinical strains of *Saccharomyces cerevisiae*, each of which contained a barcode that marked a segregant[35,36]. We then integrated a genome-scale library of 8,046 guide RNAs (gRNAs), each marked with a barcode, at an adjacent site. Pooled competitions, double barcode sequencing, and lineage trajectory analyses enabled precise measurement of the fitnesses of ~875,000 segregant-gRNA combinations. This large dataset allowed us to identify 460 genes whose perturbation has different effects across segregants and to characterize factors that cause these background effects.

## Results

### Construction of a double barcoded library of segregants with inducible CRISPRi perturbations

We previously generated, genotyped, and barcoded 822 haploid, *MATα* BYx3S *ura3Δ* segregants carrying a genomic landing pad at the neutral *YBR209W* locus[35,36] (Supplemental Figs. 1 and 2). The landing pad contains a loxP site and a partial *URA3* marker for site-directed integration of constructs and recovery of integrants by selection for *URA3*. The barcodes enable measurement of each segregant's frequency in a pool by amplicon sequencing. When measured at multiple time points, these frequencies can be used to infer the relative fitness of each segregant in a pool[34,35]. To prevent cell clumping and flocculation during pooled experiments, the main flocculin (*FLO11*)[37] and the primary transcriptional activator of cell-cell and cell-surface adhesion phenotypes (*FLO8*)[38] in *S. cerevisiae* were deleted from BY and 3S prior to mating. We randomly chose 169 barcoded BYx3S segregants from different tetrads for use in the current study (Supplemental Data 1 and 2). Most of these segregants were represented by a single barcode in the panel. To enable tests for reproducibility, 30 segregants were represented by 2–3 distinct barcodes.

To introduce genetic perturbations into the segregants at high throughput, we also designed a barcoded CRISPRi plasmid library that could be integrated into the genomic landing pad adjacent to the segregant barcode (Supplemental Figs. 3 and 4). An inducible CRISPRi system[33] was chosen because it provides several key advantages. CRISPRi perturbations are only induced during phenotyping experiments, which helps prevent accumulation of de novo suppressor mutations during strain construction and culturing. The use of inducible CRISPR interference also allows essential genes to be perturbed, expanding the space of possible target genes. Finally, CRISPRi-mediated gene repression is orthogonal to a cell's endogenous transcriptional machinery and should perform similarly across strains. Each plasmid in this library contained an endonuclease-dead Cas9 (dCas9) fused with the Mxi1 repressor domain, an inducible single guide RNA (gRNA) targeting dCas9 to a specific promoter, a unique 20-nucleotide barcode, a loxP site to enable chromosomal integration,

and the remaining portion of *URA3*. This plasmid library targeted a total of 1739 genes that have been reported to be essential under fermentative or respiratory growth conditions with 8,864 unique gRNAs, with an average of five distinct gRNAs per gene[34] (Supplemental Data 3, Supplemental Fig. 5). 91 unique gRNAs targeting intergenic and noncoding regions rather than promoters were also included with the intent of serving as controls. Roughly 10 distinct barcodes were generated for each gRNA on average, resulting in an estimated plasmid library size of ≥90,000. All gRNAs were under the control of a tetO-modified *RPR1* promoter, enabling gRNA induction by the addition of anhydrotetracycline (ATC) to liquid growth medium. ATC itself has no effect on yeast growth at the concentrations used in this assay[33,39,40].

We integrated this library of barcoded CRISPRi plasmids into each segregant and the BY parent using large-scale transformations and Cre/loxP-mediated site-specific recombination. Natural variation in transformability impeded integration of the library into certain segregants, as well as into the 3S parent (Supplemental Fig. 6). Based on colony counts and amplicon sequencing of double barcodes, we estimated that ~10,000–15,000 integrants, or ~1.1–1.7 barcodes per gRNA, were recovered per successfully transformed strain (Supplemental Data 4). Transformants were grown to stationary phase and roughly $10^8$–$10^9$ cells per strain were combined into an initial pool, which was used as the initial time point (T0) in the subsequent fitness assays.

### Phenotyping of the segregant-gRNA pool

Plasmids were designed so that after integration, chromosomally-encoded segregant and plasmid barcodes would be 99 bp apart, enabling their co-amplification by PCR and sequencing within the same paired-end read (Fig. 1a, Supplemental Fig. 3). Unlike previous experiments involving single barcodes[35], this double-barcode design allows both the genotype and its integrated gRNA to be identified in a single sequencing read. Sequencing of barcode pairs revealed that 169 unique segregants (223 segregant barcodes) and 8046 unique gRNAs (107,141 gRNA barcodes) targeting 1721 distinct genes were present in T0. Each gRNA appeared in 110 segregants on average, with T0 containing ~2,800,000 double barcodes in total. On average, ~4.5 gRNAs targeted a given gene (Supplemental Data 5).

The T0 pool was split into three separate fitness assays: two replicate experimental assays where the gRNA expression was induced with ATC (ATC1 and ATC2, experimental conditions) and one control assay where no ATC was added and gRNA expression was not induced (CON, control condition). All assays were done in synthetic complete medium containing glucose as the fermentable carbon source and lacking uracil, which maintained selection for the *URA3* marker generated by integration of CRISPRi plasmids. We performed the assays for 10 generations in serial batch culture, diluting 1:4 roughly every two generations, with a bottleneck size of ~$2.5 \times 10^{10}$ cells. We sequenced each assay at zero, two, four, six, and 10 generations (T0, T1, T2, T3, and T5). At each of these time points, all double barcodes were extracted, sequenced, and counted. For each double barcode, frequency measurements across all five time points were combined into a single lineage, using Bartender clustering to account for sequencing errors and mutations[41]. These lineages were used to estimate the relative fitnesses of segregant-gRNA combinations using PyFitSeq[42] (Fig. 1b, Supplemental Figs. 7–10; Supplemental Table 1; Supplemental Data 6–8). PyFitSeq estimates the fitnesses of lineages relative to the mean fitness of the whole population. Specifically, PyFitSeq models the fitness of each individual lineage as constant across time and allows the mean population fitness to vary over time with changes in the frequencies of different lineages[42]. Because fitness estimates are relative to the mean population fitness within an assay, comparing fitnesses between assays requires a normalization step. To enable comparisons between assays, we normalized the fitnesses within each assay relative to the fitnesses of control gRNAs, which target noncoding and

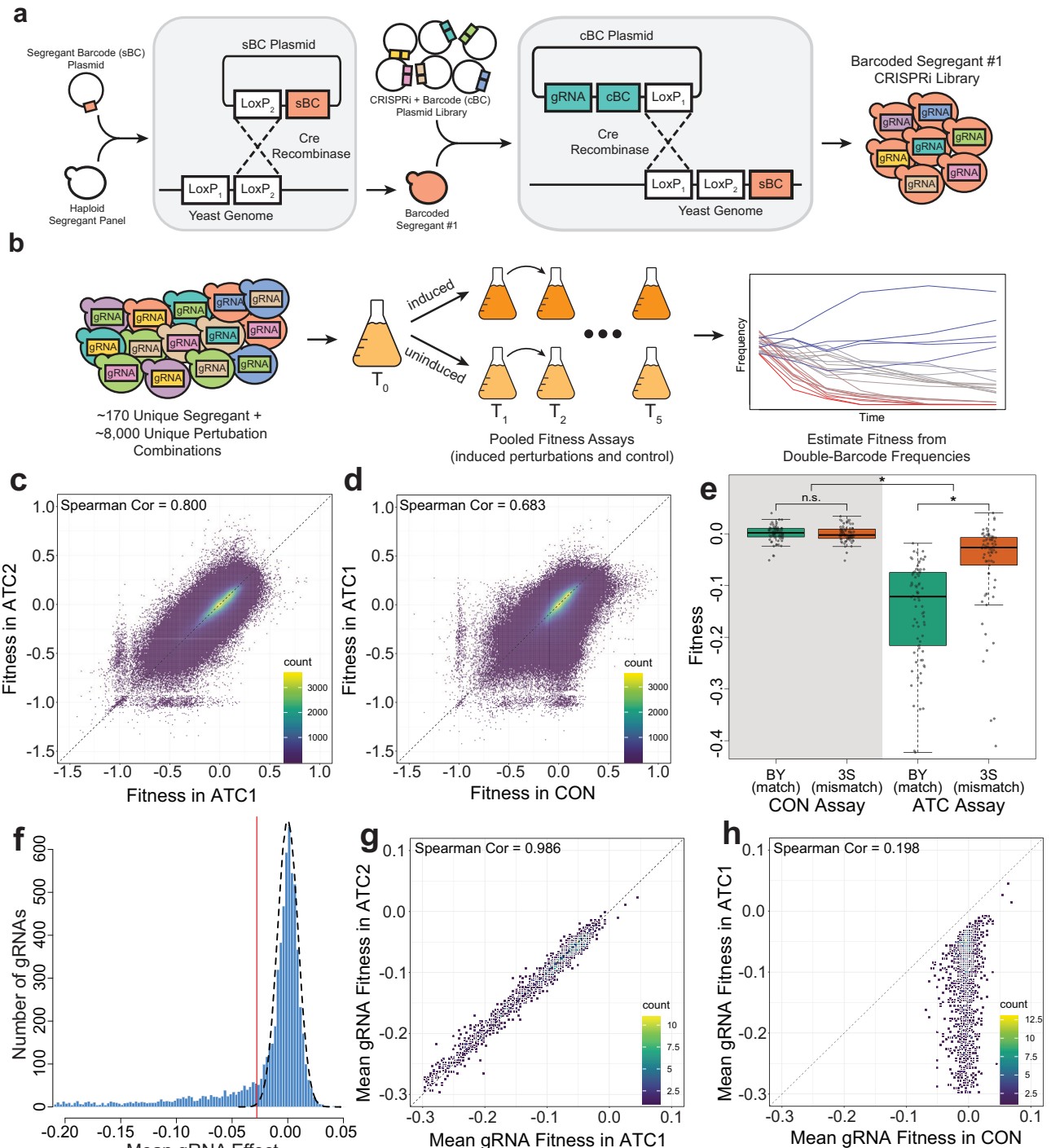

**Fig. 1 | Generating and phenotyping a panel of double-barcoded segregants carrying CRISPRi perturbations. a** Segregant-specific barcodes were first integrated into the genomic landing pad, followed by barcoded CRISPRi constructs. This resulted in a double barcode within each cell that uniquely identified its specific segregant-gRNA combination. **b** Pooled fitness assays were performed. Fitnesses were estimated using double barcode amplicon sequencing data from multiple time points. Density plots showing the fitness of each double-barcode lineage across either ATC1 and ATC2 (**c**) or CON and ATC1 (**d**). ATC1 and ATC2 are two replicate experimental fitness assays where gRNAs were induced. CON is the control assay where gRNAs were not induced. **e** Average fitness values of 74 gRNAs that directly overlap polymorphisms between BY and 3S, across both experimental and control assays. 3S alleles at binding sites have imperfect matches with gRNAs. Boxes indicate interquartile ranges (IQRs), with medians shown by horizontal lines. Whiskers indicate a distance of 1.5 IQRs from the upper or lower quartile, up to the minimum or maximum values. Asterisks and 'n.s.' respectively indicate $p < 1 \times 10^{-5}$ and >0.05 via two-sided $t$ test ($p$ value for ATC match vs ATC mismatch = $9.3 \times 10^{-09}$, $p$ value for CON match vs CON mismatch = 0.837, $p$ value for ATC vs CON = $3.0 \times 10^{-24}$). **f** Distribution of all mean gRNA effects as calculated by the mixed effects linear model. The estimated null distribution is shown by the dashed line. The threshold used for calling efficacious gRNAs is three standard deviations below the mean of this null distribution (vertical red line). The gRNA effect here is a mean calculated by averaging across all segregants carrying the gRNA. Density plots showing the average fitness values for each gRNA across either ATC1 and ATC2 (**g**) or CON and ATC1 (**h**). For each gRNA, replicate barcodes within each genotype were averaged before taking the mean across all genotypes. In all density plots, the data points–double barcode lineages in (**c**) and (**d**) and gRNAs in (**g**) and (**h**)–are organized in bins to generate heatmaps.

intergenic DNA. We also adjusted all three assays such that the mean fitness of all lineages in the CON assay was zero.

Because all three fitness assays contain the same double barcode lineages, we can directly compare the fitnesses of lineages between assays. We expected that lineages would have highly correlated fitnesses between the two replicate ATC assays. In part, we had this expectation because we previously showed these segregants have different fitnesses[35,36], implying that a major source of variance in fitness in this study should be baseline fitness differences among segregants. Additionally, we expected that lineages carrying efficacious gRNAs would have different fitnesses in ATC and CON assays. However, only a subset of the gRNAs have potent biological effects under our assay conditions[33,34], so we expected that only some lineages would exhibit differences between ATC and CON. Most lineages should thus correlate well between ATC and CON due to the lack of gRNA effects, with the few lineages carrying efficacious gRNAs showing large deviations. Indeed, we found that fitness estimates for the two replicate ATC assays were highly correlated (Fig. 1c, Spearman's correlation: 0.8), and the correlation of fitness estimates between CON and ATC1 was only slightly lower (Fig. 1d, Spearman's correlation: 0.683). As expected, a subset of lineages showed decreased fitness in ATC1 compared to CON, presumably because these lineages carry gRNAs with significant fitness effects.

To identify specific efficacious gRNAs, we used mixed effects linear models that accounted for both the baseline fitnesses of segregants and the mean fitness effects of gRNAs across segregants. For the remainder of this paper, we use the 'mean effect' of a gRNA to refer to its average fitness effect across all segregants, as estimated from a mixed effects linear model. Similar to a prior study using the same gRNA library[33], ~33% of gRNAs (2290) had significant mean effects. The specific gRNAs that were efficacious in our experiment showed substantial overlap with those identified in the prior study; of 731 gRNAs previously shown to have an effect, 89% (651) also had significant mean effects in our data set. To confirm that our technique was capable of detecting gRNAs with variable effects across segregants, we examined gRNAs in our library that directly target polymorphic sites. Because these gRNAs are all specific to the BY genome, a 3S allele at the binding site is likely to reduce, if not eliminate, gRNA binding and efficacy. For these gRNAs, we found that segregants with the 3S allele at the binding site had significantly higher fitness than segregants with the BY allele, implying that gRNA binding was impacted by the SNP. This difference in fitness was correlated with the distance between the disrupting SNP and the protospacer adjacent motif, with smaller distances leading to a larger difference between BY and 3S alleles (Supplemental Fig. 11). Additionally, in the CON assay, where gRNAs are not induced, this difference was not significant (Fig. 1e, Supplemental Fig. 11 and Supplemental Table 2). This implies that the fitness differences observed in our assays were due to the biological effects of the gRNAs, and that our technique had sufficient precision to detect differences in gRNA activity across segregants. All gRNAs that bind near polymorphic sites were excluded from further analysis. As an additional validation, within the same assay, efficacious gRNAs targeting the same gene had correlated effect sizes (Pearson correlation = 0.382, $p = 5.6 \times 10^{-19}$; Supplemental Fig. 12). Such a correlation was never observed across 1000 permutations of the same data (Pearson correlations in permutations: mean = $5.4 \times 10^{-4}$, min = −0.143, max = 0.118; Supplemental Fig. 12).

Because all genes targeted in this study have been previously annotated as essential under fermentative or respiratory growth conditions[34], we did not expect gRNAs to show positive effects. Thus, in addition to the ANOVA model, we employed a conservative effect size threshold to identify efficacious gRNAs. We only considered a gRNA further if its mean effect was at least three standard deviations below the mean of a null distribution inferred from the data (Fig. 1f, Supplemental Data 9). Of the 6977 CRISPRi perturbations in this study that had invariant binding sites, 1536 met all of these criteria, targeting

787 distinct genes (Supplemental Fig. 13). To validate this thresholding step, a single fitness estimate was obtained for each of these efficacious gRNAs by averaging all of its genotype and gRNA barcode replicates within an experiment. These average fitnesses were highly correlated between ATC1 and ATC2 (Fig. 1g, Spearman's correlation: 0.986), but not between ATC1 and CON (Fig. 1h, Spearman's correlation: 0.198).

A limitation of our method is that it may take time for lineages harboring efficacious gRNAs to equilibrate after induction. To investigate this potential time dependence, we re-calculated both fitness and mean gRNA effect estimates using only the final three time points from the fitness assays. Mean gRNA effects were higher when only these 'late' time points were used (Supplemental Fig. 14). However, the fitness estimates from late time points were also less reproducible for lineages that dropped to low frequency over time, which should occur if an efficacious gRNA is present (Supplemental Fig. 15). When all time points were employed, as was done throughout the paper, mean gRNA effect estimates remained highly correlated with estimates from late time points (Spearman's correlation = 0.804), and fitness estimates were more reproducible (Supplemental Fig. 15, Supplemental Table 3).

## Identification and characterization of background effects

We measured the fitness of each segregant-gRNA combination under both induced and uninduced conditions, making it possible to explicitly test for interactions between gRNAs and segregants using mixed effect linear models. At a False Discovery Rate of 0.05, 699 of the 1536 efficacious gRNAs had significant background effects based on a gRNA-segregant interaction term (Supplemental Data 9). These results suggest that segregants do not show major differences in global CRISPRi efficacy, as the majority of efficacious gRNAs (~54%) have indistinguishable effects across segregants. The minority of gRNAs that show background effects target 460 distinct genes in total (58% of all genes targeted by efficacious gRNAs). Prior work across model systems has shown that 15%–32% of all genes show background effects[13]. However, excluding genes whose perturbation does not show measurable effects in any individual has been shown to increase estimates of the prevalence of background effects to as high as 74%–89%[13]. Thus, the proportion of genes that we identified with background effects is in line with other studies. For the remainder of this paper, we focus on a reduced data set, selecting a single gRNA for each of the 460 genes. Selections were made based on the significance of the background effect, prioritizing gRNAs with mapped loci (as described later).

We next identified factors that caused gRNAs to show variable effects across segregants. To compare fitness effects across diverse gRNAs, we first estimated the effect of each individual gRNA in each individual segregant using mixed effects linear models. Then, the mean effect of a gRNA was subtracted from these segregant-specific values. This assigned a single 'deviation value' to each segregant, representing the difference between the effect of the gRNA in a specific segregant and its effect on average (Fig. 2a). This process was repeated for each gRNA, assigning a deviation value to every observed combination of gRNA and segregant (Fig. 2b, Supplemental Data 10). We found a slight negative relationship between the deviation values (the difference between the effect of a gRNA in an individual segregant and the mean gRNA effect) and the baseline fitness of that segregant, with the same gRNA having a more detrimental effect in higher-fitness strains (simple linear regression $R^2 = 0.149$, $p < 1.0 \times 10^{-100}$, 95% bootstrap confidence interval: 0.145–0.152; Fig. 2c). This result is consistent with prior work on the relationship between the fitness effects of deleterious mutations and baseline fitness[6]. In addition, the variance of a gRNA's deviation values was related to its mean effect size, with more detrimental gRNAs showing larger variances (simple linear regression $R^2 = 0.543$, $p = 6.1 \times 10^{-80}$, 95% bootstrap confidence interval: 0.459–0.618; Fig. 2d, Supplemental Fig. 16). However, highly detrimental gRNAs should also have higher variance as a result of increased

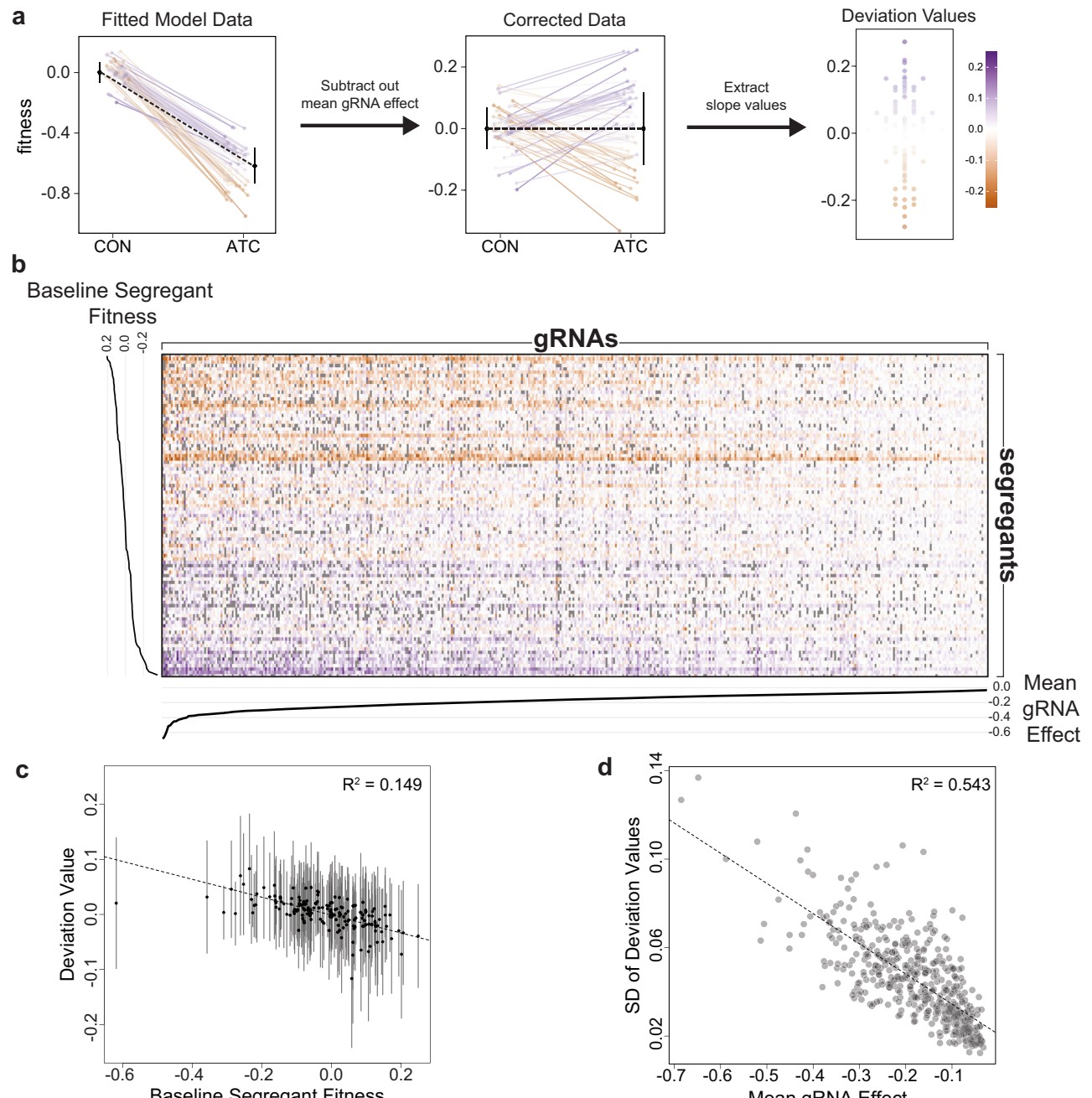

**Fig. 2 | Calculating deviation values for all segregant-gRNA combinations.**
**a** Schematic representation of how deviation values are calculated from fitness.
**b** Heatmap of all deviation values. Segregants are shown along the y-axis, ordered by mean fitness, and gRNAs are shown along the x-axis, ordered by mean effect size. Missing combinations are shown in gray. Only rows and columns with 30% or less missing data are shown. **c** Relationship between baseline segregant fitness and deviation values for that genotype ($n = 170$, including 169 segregants and the BY parent). Dots indicate average deviation value, and lines indicate 2 standard deviations. The 95% bootstrap confidence interval for the $R^2$ value of 0.149 was 0.145–0.152. **d** Relationship between mean gRNA effect and the standard deviation (SD) of deviation values for that gRNA, with the best-fit line shown. The 95% bootstrap confidence interval for the $R^2$ value of 0.543 was 0.459–0.618.

measurement noise, caused by lineages dropping to low frequency after gRNA induction. These results indicate that the baseline fitness of a segregant (without any perturbation) and the mean effect of the gRNA can both impact background effects. Baseline fitness impacts the direction of a background effect in an individual segregant and mean gRNA effect impacts the magnitude of background effects across segregants.

We also determined the extent to which genetic factors segregating in the cross explain the deviations. Using replicated measurements among genotypes and gRNAs, we estimated broad sense heritability ($H^2$), which measures the total genetic contribution to a trait[43]. $H^2$ was 0.737 on average (sd = 0.06, min = 0.54, max = 0.88, Fig. 3a), implying that background effects identified here have a mostly genetic basis.

All loci contributing to a background effect show epistasis with a perturbation (pairwise epistasis). However, some loci may interact not only with a perturbation but also each other (higher-order epistasis). The relative contribution of pairwise vs. higher-order epistasis underlying background effects can be estimated by comparing $H^2$ and narrow sense heritability ($h^2$), which measures only the additive genetic

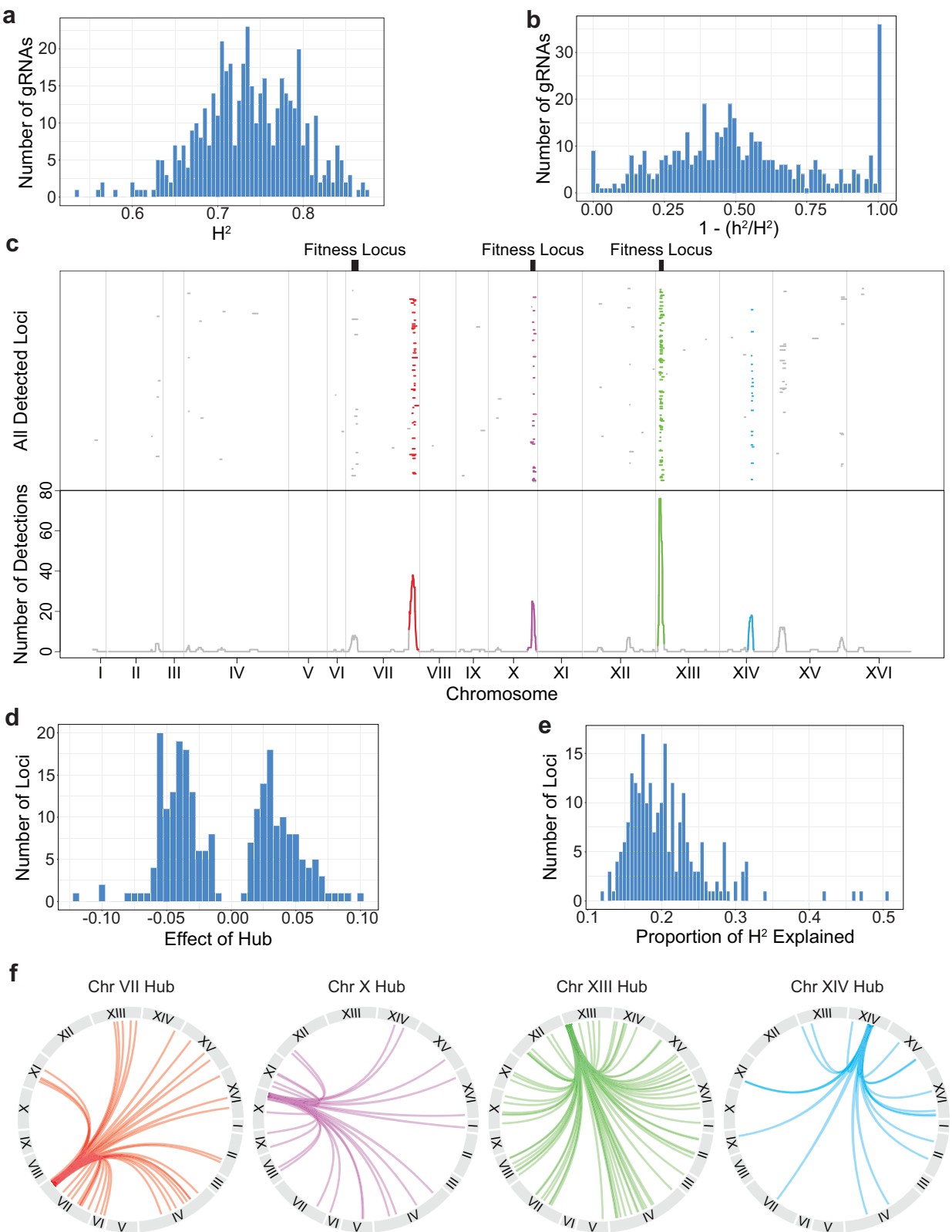

contribution to a trait[43]. Using genomic relatedness-based estimation, we found that $h^2$ was 0.358 on average (sd = 0.193, min = 0, max = 0.775). Thus, 51.4% (1−0.358/0.737) of the genetic basis of background effects could be attributed to higher-order epistasis on average (Fig. 3b). Thirty-five of 460 gRNAs (7.6%) were estimated to have zero narrow-sense heritability. A biological basis for this finding could be that the effects of these gRNAs depend on higher-order epistasis

between multiple loci and the perturbations[18–22]. However, it is also possible this finding is a technical artifact of narrow-sense heritability estimation.

## Hub loci control the phenotypic effects of many gRNAs
We next used linkage mapping to identify interacting loci that cause gRNAs to show variable effects across segregants. Specifically, we

**Fig. 3 | Using deviation values as traits in linkage mapping. a** Histogram of broad sense heritability (H²) values for all gRNAs with background effects. **b** Histogram of the ratio between narrow sense heritability (h²) and H² for all gRNAs with background effects, shown as (1 - h²/H²). **c** Linkage mapping on deviation values. Upper panel: Visualization of all 2x-log₁₀(pval) drops for the gRNAs with significant peaks. Each row is a unique gRNA. Locations of major fitness loci are indicated by black bars. Lower panel: Number of overlapping 2x-log₁₀(pval) drops at each nucleotide position along the genome. Regions where more intervals overlap than expected by chance (hubs) are shown in unique colors. Vertical lines indicate chromosome boundaries. Linkage mapping results for the replicate assay (ATC2) are available as Supplemental Fig. 17. Mapping was performed as described in the Methods section 'Linkage mapping'. **d** Histogram of all effect sizes for the detected loci. These values were obtained from the linear model used for linkage mapping, *deviations - locus*, taking the coefficient of the *locus* term. This is roughly equivalent to the difference between the mean deviation value of segregants carrying the 3S allele and the mean deviation value of segregants carrying the BY allele. **e** Histogram of the proportion of heritability explained by each detected locus, calculated as R²/H². **f** Interaction plots connecting gRNA location to the locations of interacting loci shown in Fig. 3c.

treated the deviations as trait values and mapped loci contributing to these deviations. By definition, loci identified in such scans show pairwise genetic interactions with gRNAs and play an additive role in the deviations. Because only 169 segregant genotypes were present in our data set, statistical power was limited, and we were only able to detect loci for a subset of the gRNAs. Further, genetic variants that modify chromatin accessibility near gRNA targets may impact CRISPRi efficacy at those sites. Such variants were a minor contributor to our data set: we found only 21 gRNAs with detected loci within 10 kb of their binding site and excluded these cases from subsequent analyses (Supplemental Table 4). Across the remaining gRNA-targeted genes with background effects, we detected 234 loci at a permutation-based significance threshold of -log₁₀(pval) ≥4.21 (Fig. 3c, upper panel, Supplemental Data 11). Of the targeted genes, 172 had a single mapped locus, 28 had two mapped loci, and two had three mapped loci.

Detected loci were not evenly distributed across the genome (Fig. 3c, lower panel). Across all loci, the BY allele produced higher deviation values roughly as often as the 3S allele did (Fig. 3d). On average, individual loci explained 21% of a deviation's total genetic basis (sd = 0.055, Fig. 3e). Of the 234 detected loci, 157 (67.1%) were grouped into one of four 'hubs' that interacted with a larger number of genes than expected by chance. Hubs on Chromosomes VII, X, XIII, and XIV showed linkage to 38, 25, 76, and 18 gRNA-targeted genes, respectively (Fig. 3f, Supplemental Table 5). Similar results were obtained from the replicate experimental assay (Supplemental Fig. 17). The gRNA-targeted genes to which each hub showed linkage were largely distinct, with only 7 genes connected to multiple hubs. This result further supports the notion that the segregants do not show major differences in global CRISPRi efficacy. We also found four other loci that are likely hubs but did not pass our detection threshold (Supplemental Fig. 18). Our identification of hubs is consistent with a recent study in which the fitness effects of transposon insertions were measured in a panel of yeast segregants and loci were identified that modified the effects of numerous insertions[6]. Further, two hubs (Chr XIII and XIV) and two loci that interacted with multiple gRNAs but did not pass the threshold to be called hubs (Chr XII and XV) in our study overlapped multi-hit loci found in this other study.

In addition, we mapped three loci that contribute to the baseline fitness of the segregants (Fig. 3c). This low number of detected fitness loci was due to the small number of segregants tested. The hubs on Chr X and XIII each overlapped one of these fitness loci, implying that the same locus can affect both baseline segregant fitness and the effects of genetic perturbations. We found that 43% of all detected loci with effects on deviations were mapped to these two hubs. However, as we show below, these loci had different impacts on baseline fitness and CRISPRi perturbations.

### Properties of hub loci that interact with many genetic perturbations

We found that the effect of each hub shows a different relationship with the mean effects of the gRNAs it interacts with. The Chr VII and XIV hubs exhibit the greatest effects in the presence of gRNAs that cause the largest fitness deficits. These hubs were not detected as significant loci during the linkage mapping of baseline fitness (Fig. 3c),

but were instead identified through their interactions with multiple gRNAs (Fig. 4a, b). In the presence of gRNAs interacting with either the Chr VII or XIV hub, the 3S allele of the locus becomes beneficial relative to the BY allele. This is true for all 38 of the gRNAs that interact with the Chr VII hub and all 18 of the gRNAs that interact with the Chr XIV hub. The magnitude of the difference between genotype classes at these hubs is positively related to the mean effect of a gRNA (Chr VII simple linear regression R² = 0.44, *p* = 6.2 × 10⁻⁰⁶; Chr XIV simple linear regression R² = 0.63, *p* = 7.8 × 10⁻⁰⁵).

By contrast, the Chr X and XIII hubs show the smallest effects in the presence of gRNAs that cause the largest fitness deficits. These hubs were both detected as significant loci during the linkage mapping of baseline fitness (Fig. 3c), with the 3S allele again being beneficial. However, the fitness effects of both hubs are usually attenuated in the presence of interacting gRNAs (Fig. 4c, d). This is true for 23 of 25 of the gRNAs that interact with the Chr X hub and all 76 of the gRNAs that interact with the Chr XIII hub. Unlike the Chr VII and XIV hubs, the magnitude of the difference between genotype classes at the these hubs is inversely related to the mean effect of a gRNA (Chr X simple linear regression R² = 0.52, *p* = 4.5 × 10⁻⁰⁵; Chr XIII simple linear regression R² = 0.57, *p* = 4.7 × 10⁻⁰⁸). Interactions between gRNAs and hubs can result in masking of a hub's effect (both Chr X and XIII) or even sign epistasis (Chr XIII only). These results demonstrate that genetic perturbations can quantitatively modify the effects of loci, causing them to show a spectrum of effects in an otherwise genetically identical population maintained in the same environment.

### Insights from the genetic interaction network of the yeast cell

We hypothesized that the interactions between a given hub and different gRNAs were due to a common underlying mechanism. Using the genetic interaction network from TheCellMap[44] and gRNA-targeted genes to which a hub showed linkage, we obtained multiple significant results from spatial analysis of functional enrichment (Fig. 5)[44,45]. The Chr XIII and XIV hubs were enriched for transcription (*p* ≤ 1.0 × 10⁻⁰⁴); the Chr X and XIII were enriched for mitosis (*p* ≤ 0.05); and the Chr VII and X hubs were enriched for DNA replication and repair (*p* ≤ 0.05). Several hubs also had unique enrichments: Chr VII for protein turnover and cell polarity (*p* ≤ 1.0 × 10⁻⁰⁶); Chr X for glycosylation & protein folding and MVB sorting & RIM signaling (*p* ≤ 1.0 × 10⁻⁰⁴); and Chr XIII for rRNA/ncRNA processing and vesicle traffic (*p* ≤ 1.0 × 10⁻⁰⁷). These data suggest that each hub may act through a distinct combination of cellular processes to influence the phenotypic effects of genetic perturbations. The Chr VII and XIV hubs have similar effects on mean fitness and deviation values across interacting gRNAs (Fig. 4a, b), but few functional categories are shared between their interaction networks (Fig. 5a, b). The same is true for Chr X and Chr XIII (Figs. 4c, d, 5c, d), indicating that different mechanisms can produce similar allelic relationships.

## Discussion

We developed a method for quantitatively measuring the effects of many distinct genetic perturbations in a common panel of individuals using combinatorial DNA barcoding and CRISPRi. This approach makes it possible to study interactions between distinct genetic

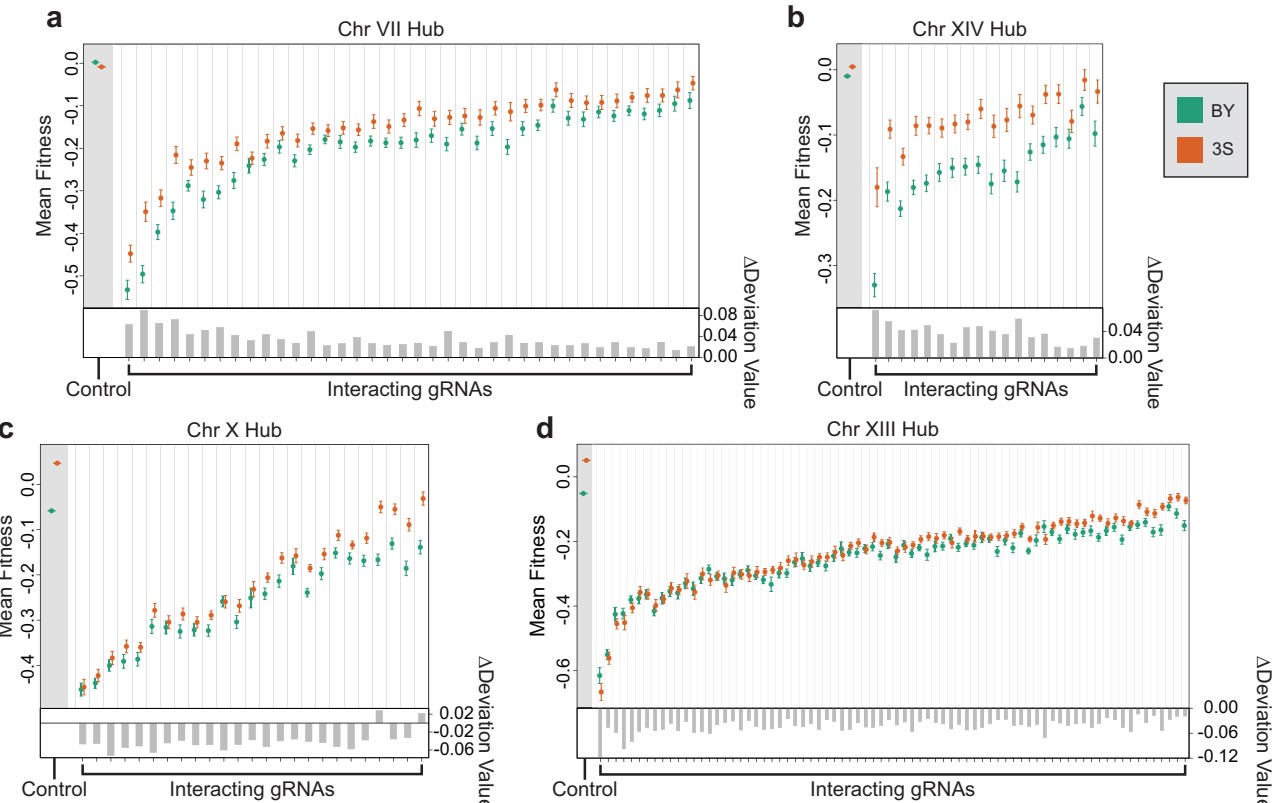

**Fig. 4 | Features of and comparisons between hubs.** Effects of the Chr VII (**a**), XIV (**b**), X (**c**), and XIII (**d**) hubs on mean fitness and deviation value across gRNAs that interact with each hub. Dot plots in upper panels: Effects of the hub on mean fitness when each interacting gRNA is induced. Control indicates a subset of 1294 gRNAs with no phenotypic effect in our data set, as determined by gRNAs with p-values greater than 0.8 for both the mean effect term and the background effect term in a mixed effects linear model (see Methods section 'Identification and quantification of gRNA effects'). Orange points show the mean fitness of all genotypes with the 3S allele at the peak marker for that gRNA's interacting locus and green points show genotypes with the BY allele. The fitnesses of replicate barcodes were averaged before calculating the mean for each genotype. Vertical bars for each point indicate standard error. The control data used the center of the hub instead of a peak marker. The gRNAs are ordered on the x-axis by the magnitude of their mean effect. Barplots in lower panels: Effects of a hub on deviation values across interacting gRNAs. The change in deviation values was calculated by taking the mean of all deviation values among genotypes with the 3S allele at the peak marker and subtracting the mean of all deviations among genotypes with the BY allele. The control data has no deviation values.

perturbations and segregating loci at the scale needed to obtain fundamental insights into background effects. Because our method requires measuring the frequencies of lineages at multiple time points and gRNAs may vary in the timing of their efficacy, a limitation is that fitness and mean gRNA effect estimates can depend on the exact time points analyzed. This means that the optimal choice of time points may need to be empirically determined based on the strains, gRNAs, and environment under study. In our experiment, we employed time points spanning the entire assay in order to maximize the reproducibility of fitness estimates. Relative to using only late time points, this choice may have led to the underestimation of some mean gRNA effects. However, the strong correlation between mean gRNA effects estimated from all and late time points suggests our findings on the relative effects of genetic perturbations across individuals are robust.

By applying our approach to 169 yeast segregants and 1536 efficacious gRNAs, we found that the effect of a genetic perturbation depends on baseline fitness (i.e., the fitness of an individual in the absence of a perturbation), the mean fitness impact of the perturbation across individuals, and the specific alleles carried by an individual at interacting loci. Our findings do not appear to be driven by major differences in global CRISPRi efficacy, as most efficacious gRNAs do not show background effects and hubs interact with different sets of gRNAs. Further, variation in chromatin accessibility at gRNA target sites cannot explain our results because the vast majority of loci that contribute to deviations did not occur near gRNA target sites. Based on the heritability explained by detected loci, we can infer that there are

likely to be multiple interacting loci contributing to each background effect. Comparison of our broad- and narrow-sense heritability estimates suggests substantial variability in the extent to which loci underlying background effects interact not only with genetic perturbations, but also each other. The large number of perturbations in our data allowed us to determine the degree to which segregating loci interact with different perturbations. While some loci interact with only one perturbation, we identified several hub loci that interact with numerous perturbations, implying that responses to many different perturbations can have a shared genetic basis.

Perturbation-locus genetic interaction networks like the one described here have been historically difficult to map because they require introducing and quantitatively phenotyping a large number of genetic perturbations in a large number of individuals. These previously unmeasured networks shape how individuals respond to different perturbations. Features of these networks likely impact how genetic background shapes the effects of perturbations, leading to incomplete penetrance, variable expressivity, and differences in robustness to perturbations. They might also control which mutations are deleterious or beneficial, thereby influencing evolutionary trajectories. Similar to more well-described networks between genetic perturbations alone[46] or between natural variation alone[35,47], perturbation-locus networks appear to contain many nodes with few interactions and few nodes with many interactions. Further study is needed to determine how these different types of genetic networks relate to each other.

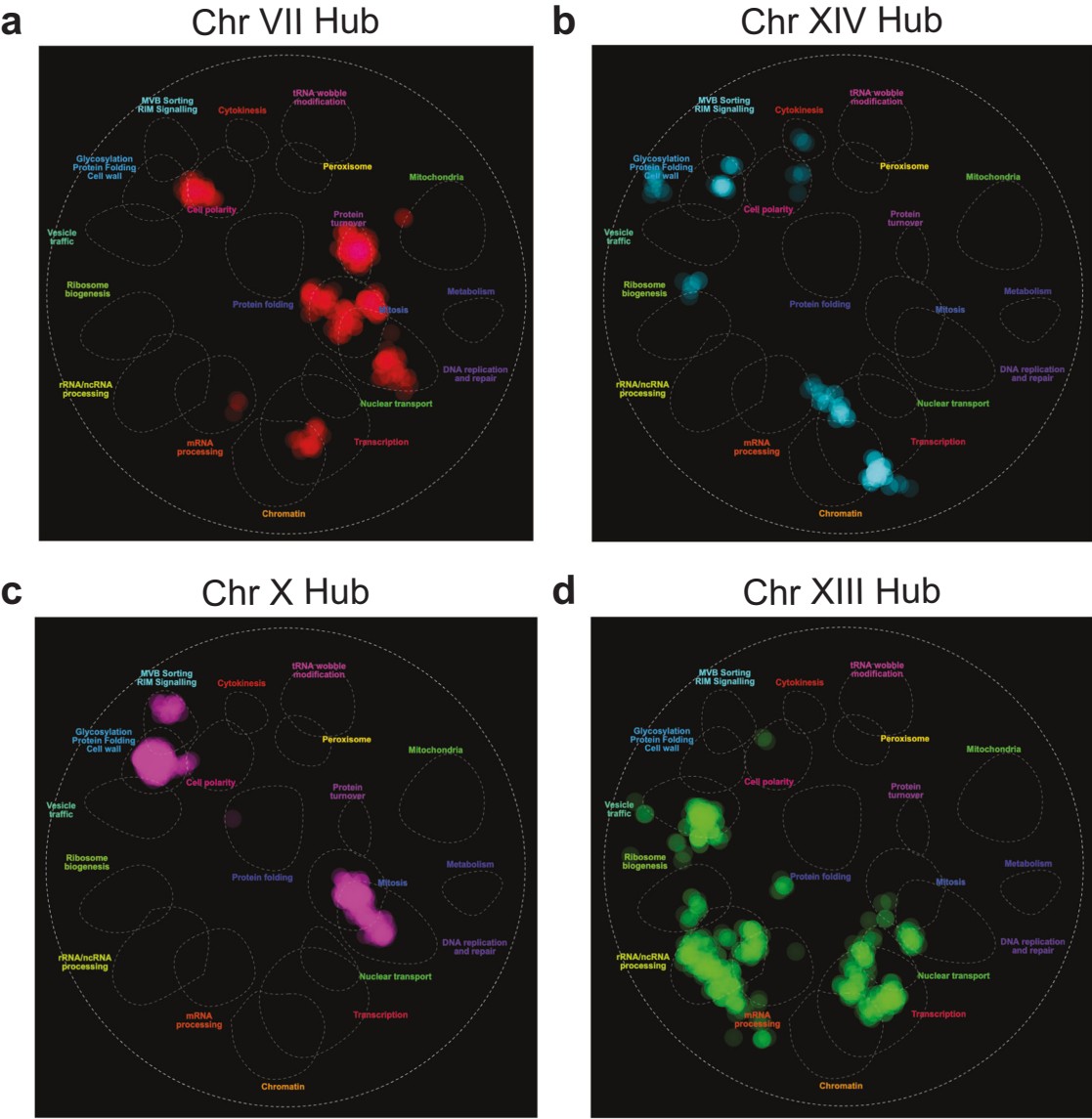

**Fig. 5 | Interaction networks for each hub.** Visualization of the interaction networks for the Chr VII (**a**), XIV (**b**), X (**c**), and XIII (**d**) hubs via spatial analysis of functional enrichment[44,45]. A list of all genes targeted by a gRNA in this hub were used as a query at thecellmap.org to create these images, at a significance of $p \leq 0.05$.

The measurable perturbation-locus network will depend on the genetic variation segregating among examined individuals, and this will necessarily be a subset of the full perturbation-locus network within a species. To simplify this problem, we focused on a systematic analysis of the perturbation-locus network between two haploid strains, BY and 3S. The current study could not fully map the network of interactions between perturbations and loci in the BYx3S cross due to the modest number of assayed segregants. This sample size was chosen to facilitate examination of a genome-scale gRNA library, with the expectation that many gRNAs would not be efficacious under our assay conditions. These results now enable follow-up studies with more segregants and fewer gRNAs. Such studies will facilitate the detection of many more loci that interact with perturbations and thereby comprehensive mapping of the network of interactions between perturbations and loci in this cross. The ultimate goal of this line of study would be to predict the outcome of a perturbation in an unseen genetic background with potential lessons for therapeutic genome editing, crop and livestock improvement, and engineering of cells and microorganisms for industrial applications.

## Methods

### Generation of barcoded haploid segregants

The barcoded BYx3S *MAT*a segregants in this study were previously reported[35,36]. In brief, all segregants were tetrad dissected from a cross between *fcy1Δ flo8Δ flo11Δ ura3Δ* versions of BY and 3S that both carried the same genomic landing pad at the neutral *YBR209W* locus. The *fcy1Δ ura3Δ* deletions provide counterselectable markers, while the *flo8Δ flo11Δ* deletions eliminate cell-cell adhesion phenotypes, such as flocculation. The genomic landing pad contains multiple partially-crippled loxP sites in close proximity, in addition to a galactose-inducible Cre recombinase. To barcode segregants, we individually transformed them with a plasmid library containing 20-nucleotide random barcodes, integrated plasmids into the landing pad by Cre/loxP-mediated recombination, and obtained random integrants. The barcodes present in the segregants, as well as segregant genotypes, were then determined via multiple Illumina HiSeq 2500 lanes[36]. Briefly, six HiSeq lanes were used to perform whole-genome sequencing on each individual segregant, with reads mapped against the S288c reference genome R64-2-1_20150113 using BWA[48]. SAMTOOLS[49] was used to convert alignments to pileups, from which high-confidence

SNP lists were generated. Segregant barcodes were identified using a separate HiSeq 2500 lane, in which the barcode was PCR amplified from each segregant using specific primers. After being isolated from sequencing reads, barcodes were clustered using Bartender[41], with clusters making up greater than 5% of all reads from a segregant being assigned as its true barcode.

## CRISPRi plasmid construction

The pE32 CRISPRi plasmid was constructed using plasmids pKR387, pKR482, and pKR919. An annotated plasmid map of pE32 is available in the supplementary information. To construct the pE32 CRISPRi plasmid, a C to T synonymous mutation was first introduced at amino acid position 1319 (Glycine, GGC - > GGT) of dCas9 on pKR387 to remove an AscI recognition site using the QuikChange II site-directed mutagenesis kit (Agilent #200523). The resulting product was transformed into NEB 10-beta cells using a standard heat shock protocol and transformants were selected on LB agar plates with 100 µg/mL of carbenicillin. From this pKR387 plasmid with the mutated dCas9, a DNA fragment containing a *GPM1* promoter, *tetR* gene, *GPM1* terminator, *TEF* promoter, dCAS9, MxiI repressor domain, and *CYC1* terminator was amplified by PCR. The resulting product was inserted into linearized pKR482 using NEBuilder HiFi DNA assembly kit (NEB E5520S) and transformed into 10-beta cells. Transformants were selected on LB agar plates with carbenicillin, and the resulting plasmid was named pSL25. This pSL25 plasmid was linearized with AscI and ClaI and ligated with a double-stranded oligonucleotide containing a partially crippled Lox66 site using T4 DNA ligase (NEB M0202S). The ligation product was transformed into 10-beta cells, selected on LB agar plates with carbenicillin, and named pSL55. Finally, a DNA fragment containing *RPR1* promoter, tetO sequence, and a hammerhead ribozyme sequence was amplified from pKR919 by PCR. The hammerhead ribozyme was included because preliminary experiments found that it improves gRNA efficacy in CRISPRi. The PCR product was inserted into pSL55, which was linearized with SpeI and BspQI, using NEBuilder HiFi DNA assembly kit. The resulting product was transformed into 10-beta, selected on LB agar plates with carbenicillin, and named pE32.

## Generation of the CRISPRi library

All CRISPRi gRNAs used in this paper were previously generated[33,34]. To generate the CRISPRi plasmid library for this paper, ~20,000 arrayed yeast strains[34], each containing a unique chromosomally-encoded gRNA, were plated onto YPD agar plates with 300 µg/mL of Hygromycin B in a 384-well format using Singer ROTOR. The yeast cells were grown overnight at 30 °C. Although these yeast strains are genetically identical except for the 20 bp gRNA, heterogeneity in growth was observed across yeast strains. To minimize bias in gRNA frequency, equal amounts of each overnight colony were transferred into 200 µL of water using the ROTOR. Cells containing essential gRNAs targeting essential and non-essential genes were then pooled separately and spun down at 3000 rpm for 15 min prior to genomic DNA extraction. Genomic DNA was extracted using MasterPure yeast DNA purification kit (Lucigen MPY80200). A DNA fragment containing a gRNA with the tracrRNA scaffold sequence, *URA3* promoter, 5′ half of *URA3*, an artificial intron, and an I-SceI cut site was amplified from the pooled genomic yeast DNA. The reverse primer used to amplify this region contained a random 20 bp sequence, which was used to mark each gRNA with an unique barcode. To minimize PCR bias, the amount of DNA required for amplification was calculated such that each gRNA is represented by ~1000 DNA molecules. A total of 330 ng and 270 ng of yeast genomic DNA was used as template for amplification of ~11,000 non-essential and ~9000 essential gRNAs, respectively. The resulting product was inserted into pE32, which was linearized with BspQI and AscI, using NEBuilder HiFi DNA assembly kit. The resulting product was transformed into 10-beta cells and selected on LB agar plates with carbenicillin. To ensure high barcode complexity, ~110,000 and ~90,000 transformation colonies were scraped and pooled prior to plasmid extraction. While the CRISPRi library used in this study initially had a library size of ~20,000 gRNAs, a previously unknown design issue with the non-essential plasmid library affected growth on selective media, resulting in substantial depletion of non-essential gRNAs in the T0 pool. As a result, all non-essential gRNAs were excluded from analyses, which had a minor impact on our total data. Raw data from non-essential gRNAs is included in the Supplementary Information.

## Linking barcodes to gRNAs in the CRISPRi library

The CRISPRi library was initially generated using fully randomized barcode sequences, necessitating a linkage step to connect known gRNAs with their barcodes. To link barcodes and gRNAs, the full plasmid library was sequenced using a PCR-free library preparation method and Oxford Nanopore MinION. The SQK-LSK109 protocol was used to prepare this sequencing library, with elution steps after bead cleanup performed at 37 °C for 10 min to increase yield. Sequencing was performed on R9.4.1 Chemistry MIN-106 flow cells. Basecalling was performed on the USC Center for Advanced Research Computing Discovery cluster. We ran Guppy v6.0.1 with the configuration dna_r9.4.1_450bps_sup.cfg on a single node with 16 threads and a V100 GPU. After basecalling, we first mapped the gRNA portion of a read to its best match among the known gRNAs and then mapped the barcode portion of read to its best match among the known gRNA barcodes from barcode sequencing of the T0 pool. Fuzzy string matching was done in Python using the process.extractOne() function in the rapidfuzz library[50].

## Transformation of haploid strains with the CRISPRi library

The strains used in this study were randomly selected from the haploid panel described above. All selected strains were individually transformed with the full CRISPRi library, using a modified version of the standard lithium acetate protocol[51] that scaled up all volumes by a factor of 10. Roughly $8.0 \times 10^8$ cells and 30 µg of plasmid library were used for each transformation. Immediately after transformation, cells were grown in 3.4 mL of YP + Galactose liquid media for 18–20 h to induce integration of the plasmid while minimizing cell division. After induction, cultures were grown in SC - URA liquid media in order to select for cells that successfully integrated the plasmid. Since SC - URA is not immediately lethal to non-integrated cells, cultures were first grown in 60 mL of selective media for 48 h. Next, 20 mL of the culture was combined with 40 mL fresh SC - URA to perform a second setback. This two-step enrichment was done to ensure that as many cells in the cultures were true integration events and to maintain the barcode and gRNA diversity among yeast transformants. After enrichment, 12.5 mL of the stationary phase culture was saved and frozen as a freezer stock.

## Fitness assay

One full freezer stock from each transformed segregant (roughly $1.25 \times 10^8$ cells) was used for the fitness assay, regardless of transformation efficiency. This value was chosen assuming a roughly uniform distribution of gRNA frequency within each segregant, leading to an expectation of $\geq 10^4$ cells per lineage. All freezer stocks were thawed and grown to stationary phase in SC - URA liquid media. Then, 10 mL from each culture (roughly $2 \times 10^9$ cells) was combined to generate a 2.5 L T0 pool. Multiple T0 cell pellets were frozen for fitness assays and DNA extraction. To begin the assay, 250 mL of the T0 pool (roughly $5 \times 10^{10}$ cells) was thawed and inoculated into 750 mL fresh SC - URA liquid media. To induce gRNA expression, ATC was added to experimental flasks (assays) at a final concentration of 250 ng/mL. ATC was not added to the control assay. To generate the first time point (T1), assays were grown for 24 h at 21 °C with shaking. The majority of the culture was harvested and saved as a frozen cell pellet, while the remaining 250 mL was inoculated into 750 mL of fresh media and grown for another 24 h. This process was repeated to generate the

remaining time points (T2 through T7). The ATC1, ATC2, and CON assays were grown, harvested, and diluted in parallel, under identical growth conditions.

## Library preparation for barcode sequencing

DNA was extracted from frozen cell pellets using the Zymo Research Quick-DNA Fungal/Bacterial Midiprep Kit, with roughly 120 μg extracted from each time point. In order to separate the barcode region from the rest of the genome, extracted DNA was digested for 18 h with the restriction enzyme I-SceI. The ~250 bp barcode region was then isolated from the genomic DNA via a standard AMPure XP bead cleanup protocol. Briefly, cleanup was performed by incubating DNA with 0.4X beads for 15 min, taking the supernatant and incubating with 0.8X beads for 15 min, then discarding the supernatant and washing DNA off the beads. 70% Ethanol washes were performed between each step. Total DNA after bead cleanup ranged from around 20 μg to 30 μg. The resulting DNA was first amplified with a 5-cycle PCR with Phusion polymerase, using 200 ng of input DNA and primers specific to the double barcode region. These primers were:

1. AATGATACGGCGACCACCGAGATCTACACNNXXXXNNACACTC TTTCCCTACACGAC
2. CAAGCAGAAGACGGCATACGAGATNNXXXXNNGTGACTGGAGT TCAGACGTGTGCTCTTCCGATCT

The N's in the above sequence represent the random unique molecular identifiers (UMIs) used to account for and eliminate PCR duplicates that could appear in later amplification steps. The X's represent Illumina multiplexing indices used to enable eventual pooling of multiple libraries onto a single sequencing lane. To ensure consistency of the PCR protocol, each library was split in half on this step, with each aliquot amplified with a different set of multiplexing indices. Final results and data output were similar across all index combinations for each library.

After the first PCR step, all products were pooled together and purified with Zymo Research DNA Clean & Concentrator-5 columns, using 150 μL of PCR product per column and eluting into 30 μL. The purified product was used to set up a second 26-cycle PCR with Phusion polymerase, using 15 μL of template. Universal Illumina primers P1 and P2 were used for this reaction. All PCR products from this reaction were pooled and put through another Clean & Concentrator-5 column, with 375 μL of product per column, and eluted into 35 μL. The cleaned product was re-pooled, and the 200–300 bp region was extracted via agarose gel electrophoresis. DNA was recovered from the gel using Zymo Research Zymoclean Gel DNA Recovery Kits and Clean & Concentrator-5 columns. The purity and concentration of the final library was assessed using Life Technologies Qubit 2.0 and Thermo Fisher NanoDrop ND-1000 Spectrophotometer. The structure, sequence, and diversity of each library was verified via Sanger sequencing.

## Barcode sequencing

150 bp paired-end sequencing was performed for all time points, on either a NovaSeq 6000 or a HiSeq 4000 platform. Up to three multiplexed time points were included on each NovaSeq lane, with each time point being allocated roughly 800 million reads regardless of platform. Each sequencing lane was performed with 25% PhiX spike-in to ensure nucleotide diversity during the run, since the majority of the amplicons consisted of fixed bases. Sequencing data was analyzed with custom-written code in Python and R. For the initial processing of large files, USC's Center for Advanced Research Computing Discovery cluster was used. Reads were sorted based on several multiplexing indices unique to each time point. Forward reads were used to extract gRNA barcodes and reverse reads were used for genotype barcodes, with reads filtered out if the expected fixed nucleotides did not appear immediately downstream of either barcode. These downstream

sequences are CCCGAGTCGCGATAA and TACCGTTCGTATAGG for the gRNA and segregant barcodes, respectively. Reads were also filtered out if the first 35 bases of either the forward or reverse read had an average Illumina quality score below 30.

Consistent with previous publications[30,31,35], very similar barcodes were clustered together and treated as the same sequence in order to minimize the effects of sequencing errors and random mutations on barcode counts. Clustering was performed using both Bartender[41] software and the rapidfuzz package (v2.13.0)[50], and was done separately for genotype and gRNA barcodes. First, a list was generated of every unique barcode detected across all time points by the extraction method described above. Consensus sequences were identified from this input using Bartender, with the merging threshold disabled (-z -1) to avoid frequency-based clustering. Each consensus sequence was then compared against the entire list of verified reference barcodes using rapidfuzz, with similarity scores of 90 or higher considered a match. This threshold is roughly equivalent to a 1–2 nucleotide difference between the Bartender consensus sequence and the known reference barcode. The majority of reads clustered and matched this way (>95%) were associated with a perfectly-matched reference barcode (score = 100). Reference lists for genotype barcodes were obtained through separate HiSeq 4000 lanes on pooled segregant barcodes[36], in addition to Sanger sequencing on individual segregants. Reference lists for gRNA barcodes were obtained from separate Nanopore sequencing lanes (see section 'Plasmid Library Preparation' for more detail).

After clustering, the frequency of each double barcode in each time point was calculated. Barcode amplicons undergo PCR before being sequenced, and PCR duplicates can cause significant bias if they are not removed when calculating frequencies. To account for this, a random 4-mer UMI was added to every multiplexing index during the library preparation, for a total of 8 random nucleotides per fragment. These UMIs, which are added on an early step of PCR, are distinct from the 20-nucleotide barcodes described above, and are only used to eliminate PCR duplicates. Because the total number of possible 8-nucleotide combinations ($4^8$ = 65,536) is significantly higher than the expected number of reads for any individual double-barcode fragment, it is unlikely that the same DNA template will contain the same UMIs by chance. This can facilitate estimation of barcode frequencies by counting unique UMIs detected for each double barcode. However, UMI-containing primers were not fully removed from the sample by PCR cleanup kits due to their large size (>50 bp), causing unique UMIs to be added in all stages of PCR, not just the first few cycles. This meant that not every PCR duplicate could be identified via UMIs and removed from our frequency measurements.

## Removing PCR chimeras

The double barcode constructs in this study consist of two highly variable barcodes separated by a region of invariant nucleotides, which can enable the production of spurious double barcodes (chimeras) during PCR[52]. To quantify and account for chimeras, we chose ~10 individual segregants and extracted their barcodes in isolation from each other. These individually-prepared segregants were sequenced via the methods described above on the HiSeq 4000 platform. Since segregants were completely isolated during this library preparation, there was no opportunity for PCR chimeras to be generated. This allowed us to construct a high-confidence list of the true gRNA barcodes present in those 10 segregants, which in turn identified double barcodes from the fitness assay data that were likely chimeras. The number of double barcodes per segregant identified as chimeric based on invalid combinations of barcodes from this data was consistently <10%. To more broadly correct these data for chimeras, a linear model was fit for the frequency of each known chimera as a function of the total frequency of the corresponding segregant and gRNA barcodes in that sequencing lane: *(frequency of chimera) ~ (total frequency of*

*segregant barcode) + (total frequency of gRNA barcode)*. This linear model was fit separately in four early time points from the fitness assay, and the coefficients were averaged across the four models. The final model was used to correct every double barcode's calculated frequency in each time point for its expected number of chimeras, with corrected double barcode frequencies set to a minimum of zero.

## Fitness estimation

Fitness estimation was performed as previously reported[42], with minor edits to adapt the method to our fitness assay. Specifically, differences in coverage between time points (Supplemental Table 1) were first accounted for by normalizing read counts. This was done by dividing each read count by the total number of reads from that time point. All read counts were then multiplied by the same arbitrary value (450,000,000, roughly the number of reads present in T0). PyFitSeq[42] was used to estimate fitness of each double barcode lineage, which consisted of normalized read counts from five separate time points. PyFitSeq is commonly used software for fitness estimation in barcode sequencing studies[42]. This software infers the fitnesses of lineages relative to the whole population based on changes in lineage frequencies over time, to determine the fitness of each lineage relative to the mean of the population at T0. The mean population fitness is allowed to vary over time as different lineages proliferate or decline in frequency, but the fitness of each individual lineage is fixed across all time points. Fitness estimates are used to project lineage trajectories, these projected trajectories are compared to observed trajectories, and are adjusted over the course of multiple iterations until an optimum is reached. The maximum number of iterations was set to 200, and all default PyFitSeq settings were used otherwise. Lineages were removed from the data set if their log likelihood score was one interquartile range below the first quartile, or if <5 reads were present at the first time point (T0), as both cases led to inaccurate fitness estimates. Additionally, segregants with <100 associated gRNAs were removed from the data set entirely, as were gRNAs with <2 associated segregants.

PyFitSeq generates fitness estimates that are relative to the mean population fitness of each assay at T0, so any small differences between the mean fitness of each assay must be corrected via normalization. This allows the fitness estimates to be compared between different assays, and is done by selecting a subset of lineages that is expected to have the same fitness in all three assays. This normalization was performed using 83 control gRNAs that target intergenic and noncoding regions as the common reference point across the three assays. For each of the control gRNAs, the average fitness of every lineage carrying that gRNA was determined. Six of the 83 control gRNAs were removed at this step, as their average fitness was below zero and they had inconsistent effects between the experimental and control assays. After taking the average fitness of every gRNA, the median of these values was used to normalize all three assays. Specifically, the difference in these medians between ATC1 and CON was subtracted from the fitness of every lineage in ATC1, and this process was repeated for ATC2. Normalization was then performed by taking the difference in median fitness among the neutral gRNAs between ATC and CON assays, with both ATC assays normalized to CON.

To ensure the validity of this normalization method, two other common reference points were chosen. Both of these leveraged the fact that we do not expect any gRNAs to have positive effects in our data set. The first additional reference point was the set of all gRNAs whose average fitness was higher in both experimental assays than in the control assay. The second additional reference point was the set of all gRNAs whose average fitness appeared in the 4th quartile of all three assays. Normalizing with respect to both of these additional reference points produced similar values to normalizing via the control gRNAs.

Finally, to aid in interpretation of the fitness values, we adjusted all fitness estimates in all assays by the same value, such that the mean fitness of all lineages in CON was exactly zero. This was done to ensure that neutral lineages with no major fitness effects were as close as possible to an intuitive fitness value of zero, which is not produced by PyFitSeq by default. Because our analysis only concerns comparisons between fitness values, shifting the entire data set in this way has no impact on the following analysis.

## Identification and quantification of gRNA effects

After assays were normalized, mixed effects linear models were used to determine if gRNAs had a significant mean effect across all segregants, and if any background-specific effects were present. Here, we use 'mean effect' to refer to the impact a gRNA has on fitness on average, when considering all genotypes together. This is distinct from the genotype-specific deviation values discussed elsewhere in this study. The mean effect of each gRNA was determined by first identifying the subset of all lineages carrying that gRNA, then comparing the fitnesses of these lineages across one experimental assay (ATC1) and the control assay (CON) using mixed effects models. The mixed effects model *fitness ~ segregant + gRNA + error* was fit to this subset of lineages using the lme() function in the nlme package in R (v3.1.160)[53]. Here, the *segregant* term was treated as a random effect and the *gRNA* term was treated as a fixed effect. All lineages from the control assay were assumed to lack the perturbation (*gRNA* = 0), and all lineages from the experimental assay were assumed to have the perturbation present (*gRNA* = 1). If the addition of a *genotype:gRNA* interaction term significantly improved model fit, the model *fitness ~ segregant + gRNA + segregant:gRNA + error* was used for that gRNA instead. Regardless of model, if the *p* value for the *gRNA* term was significant after Benjamini-Hochberg multiple testing correction, the perturbation was considered to have a potential mean effect. Among the gRNAs with mean effects, if the *p* value for the *gRNA:genotype* interaction term was also significant after Benjamini-Hochberg multiple testing correction, the gRNA was considered to have a background-dependent effect. This was repeated for each individual gRNA. After all gRNAs had been tested for effects, this process was repeated using the second replicate experimental assay (ATC2) rather than ATC1.

Mean gRNA effects were estimated by extracting the *gRNA* term coefficient from the appropriate linear model described above. This was done for all gRNAs, regardless of their effects. Analysis of the control data indicated that some gRNAs had leaky expression, having similar mean effects across both experimental and control assays. The linear models described above also removed these gRNAs from the data set, as the mean gRNA effect was determined by comparing the same gRNA between ATC and CON assays. We found that the *p* value threshold described above was not sufficiently stringent to identify efficacious gRNAs. To set a conservative threshold for efficacious gRNAs, we took the subset of gRNAs with neutral or beneficial effects and simulated a normal distribution from this data. Perturbations were only considered to have a mean effect if they were also at least three standard deviations below the median of this simulated distribution.

In the case that a background-specific effect was detected using the method described above, segregant-specific model coefficients were extracted using the predict() function in R. This provided an estimated gRNA effect for every segregant in which that gRNA occurs. The overall mean gRNA effect was then subtracted from these values, giving us segregant-specific deviation values. These represent the direction and magnitude of each segregant's response to a gRNA after correcting out the gRNA's mean effect. Segregants with insufficient data to obtain a coefficient were excluded from this calculation. gRNAs with deviation values for less than 35 segregants (~20%) were excluded from further analysis.

## Linkage mapping

Linkage mapping was performed individually for each gRNA using deviation values as phenotypes. For each gRNA, we performed genome-wide scans using the lm() function in R, with the model lm(*deviations ~ locus*). Significance thresholds were determined by running 1000 permutations in which a random gRNA was chosen, its deviations were randomly shuffled, and the lowest $p$ value was saved. From these 1000 $p$ values, the 5th percentile was used as the significance cutoff for the respective linkage mapping model. Peak detection was performed by taking 2x-$\log_{10}(p$ value) drops for each locus that was both above the significance cutoff and at least 100 kb away from any other 2x-$\log_{10}(p$ value) drop. Any loci for which the gRNA's binding location was <10 kb away from the peak marker were excluded from analysis. These could potentially represent polymorphisms that disrupt a gRNA target site or that impact gRNA binding by modifying local chromatin accessibility. In total, 655 gRNAs were removed this way. Thus, local variation in chromatin did not have a major impact on our study.

## Heritability estimation

Reproducibility of gRNA effects across ATC1 and CON assays was used to estimate broad-sense heritability. This was possible due to the replication present in the data set, with most gRNAs and many segregants each represented by multiple distinct barcodes. First, a mean fitness estimate for each segregant was obtained from the CON assay and control gRNAs. Then, for each gRNA, these segregant fitnesses were subtracted from each corresponding double barcode fitnesses from the ATC1 assay, providing a gRNA effect estimate for every individual double barcode. These estimates were used to fit the linear model *guide_effect ~ genotype*. Using this model, broad-sense heritability for each gRNA was determined by taking the sum of squares between genotypes and dividing by the total sum of squares. Narrow-sense heritability was estimated using the sommer package in R (v4.3.2)[54]. The A.mat() function was used to generate an additive relationship matrix from the deviation values previously generated for each segregant-gRNA combination, and the mmer() and vpredict() functions were used to generate the narrow-sense heritability estimate from this matrix.

## Analysis of hub loci

A significance threshold for detecting hubs was calculated by assuming a random distribution of intervals across the genome[55]. Specifically, the genome was split into 20 kb bins and a threshold was set at 0.05/(number of bins). A Poisson distribution was used to determine the number of overlapping intervals per bin that would result in a $p$ value below this threshold. For this Poisson distribution, lambda was set as (total length of all detected 2x-$\log_{10}(p$ value) drops in bp)/(total length of the yeast genome in bp). Hubs were defined as adjacent loci with counts (number of overlapping intervals) above this threshold. The locus with the highest count was selected as the marker for that hub. In the case of ties, the genomic positions of the tied loci were averaged, and the locus closest to the average was used as the marker. All gRNA-targeted genes whose 2x-$\log_{10}(p$ value) drops overlapped this marker were considered a part of that hub. The effect of each hub locus on deviation value in the context of interacting gRNAs was extracted from the linear model used for linkage mapping lm(*deviations ~ locus*), taking the coefficient of the *locus* term. Proportion of broad-sense heritability explained by the hub locus for each gRNA was also calculated this way, by taking the variance explained by the model and dividing it by the broad-sense heritability estimate for that gRNA. TheCellMap analysis was performed by uploading lists of genes comprising each hub to thecellmap.org and downloading enrichment tables and visualizations[44]. Data visualization was performed using the circlize (v0.4.15)[56] and ggplot2 (v3.4.3)[57] packages in R.

## Reporting summary

Further information on research design is available in the Nature Portfolio Reporting Summary linked to this article.

## Data availability

The barcode sequencing data generated in this study have been deposited in the NCBI Sequence Read Archive Bioproject under accession code PRJNA986287. Data used to generate figures are available through the Source Data file. Supplemental data files 1–14 are available both as Supplementary data files and through Figshare at: https://figshare.com/s/c4c90d476ae701a6cf16. The S288c reference genome data used in this study is available at http://sgd-archive.yeastgenome.org/sequence/S288C_reference/. The data used for spatial analysis of functional enrichment via TheCellMap is available at https://thecellmap.org/. Source data are provided with this paper.

## Code availability

All code used for data processing and analysis is available through Figshare at: https://figshare.com/s/c4c90d476ae701a6cf16.

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

## Acknowledgements

We thank Cara Hull, Zach Krieger, and Daniel Lusk for feedback on a manuscript draft. This work was supported by startup funds from the University of Southern California to I.M.E., grant R35GM130381 from the National Institutes of Health to I.M.E., and grants R01AI164530 and R01HG010378 from the National Institutes of Health to S.F.L.

## Author contributions

J.J.H., T.M., I.G., M.N.M., K.R.R., S.F.L., and I.M.E. conceptualized this project. T.M., K.R.R., L.M.S., and S.F.L. generated the gRNA libraries. J.J.H., T.M., I.G., M.N.M., K.R.R., C.N.V., C.W., and T.R. performed the experiments. J.J.H. and I.M.E. analyzed the data. D.M. provided modified Fit-Seq code. J.J.H. generated the figures. J.J.H., T.M., I.G., S.F.L., and I.M.E. wrote and edited the manuscript.

## Competing interests

The authors declare no competing interests.
