## [Peer Review File · Nature Communications]

Genome-scale analysis of interactions between genetic perturbations and natural variationREVIEWER COMMENTS

Reviewer #1 (Remarks to the Author):

Hale et al use a powerful double-barcoding system that they previously developed to systematically survey genetic interactions between natural genetic variation and CRISRPi gene knockdowns (KD). They measure the fitness effects of KDs of 1,721 genes in a panel of 169 yeast segregants and identify over 400 genes whose KDs display genetic-background dependence. Furthermore, they identify four "hub" loci where the natural genetic variation interacts with many KDs, and they characterize these interactions.

Although this paper does report anything particularly unexpected, it (along with other recent papers by the Ian Ehrenreich's and Hunter Fraser's groups) sets a new standard for the field. The barcoding and CRISPR-based genetic engineering techniques developed by the authors generate remarkable amounts of high-quality data which will undoubtedly serve as an important resource for the community. However, we identified several important methodological concerns which we think the authors should address before publication.

1. We are concerned about the fact that the mean fitness of engineered strains in CON is significantly higher than zero (Figure 1H). Our understanding is that the introduction of the guide construct should have no fitness effect in the CON condition, at least in principle. Related to this, the strong correlation between fitness in CON and ATC (Figure 1d) is also surprising to us. The current explanation that this correlation arises "because only a subset of gRNAs have strong effects on fitness" does not make sense to us. To understand this better, we would have liked to see the correlation between two replicates in CON, if available. Again, we would expect no correlation between CON1 and CON2 replicates or between CON1 and ATC, if the introduction of the guide construct has no effect on fitness.

Can the authors offer some explanations for these somewhat strange observations? We can think of several potential explanations. (a) The authors measured fitness with respect to a low-fitness reference strain. (We did not understand what reference strain was used---see our major point 3 below.) If this is the case, it would seem appropriate to normalize this effect by setting the average fitness of all strains in CON to zero. (b) Systematic effects can come from noise in the reference strain frequency, e.g., if the reference barcodes are at low frequency. Noise in the reference introduces a systematic bias into the estimates of fitness effects of all mutations. (c) Guides are to some extent expressed even without induction. In this case, normalization with respect to the CON condition is unnecessary and potentially obscures some interesting effects. (d) The introduction of the guide construct itself introduces a sizable fitness defect and the correlation in Figure 1d arises because the authors are measuring largely this effect which is common to CON and ATC rather than the effect of the induction. We think that some additional analyses can rule out some of these possibilities, particularly (d) which is the most concerning. At the very least, we feel that the authors need to directly address these issues in the text.

2. Have the authors tested how stable their estimated fitness values are across time intervals? We are particularly concerned about such systematic variation because the CRISRPi perturbations are induced just before the assay. Several generations might be needed for the perturbed genes to be fully knocked down. If fitness estimates change over time, the authors could address this issue for example by considering only the fitness estimates derived from later time intervals.

3. It is unclear to us what the authors mean by 'Fitness'. We are confused about it in two contexts. First, when the authors discuss the fitness of segregants, it appears that this fitness is measured with respect to a common reference strain or strains, but we could not figure out what these reference strains are. Please clarify this.

Second, the authors state that there are 189 unique 'control' gRNAs which target intergenic regions. We initially thought that these were used as reference neutral mutations with respect to which fitness effects of KDs were estimated. But in the methods section on the 'quantification of gRNA effects' the authors state that they identify neutral gRNAs as all those with equal or higher fitness in ATC than control, and that these post-hoc identified 'neutral' gRNAs were used as for

normalization. So, with respect to what is the fitness effect of each KD in each segregant estimated? Can the authors explain why they did not use the known putatively neutral 'control' gRNAs for normalization instead?

MINOR POINTS

1. In the future, please add page and line numbers to make the lives of reviewers a bit easier.
2. "We only considered a gRNA further if its fitness effect was at least three standard deviations below a null distribution inferred from the data". Do the authors mean "below the mean of the null distribution"? It would be good to show this threshold in Figure 1F. There is a vertical line in Figure 1F, but we were not sure if this is actually the threshold discussed in the text because it seems to be less than 3 stds below the mean of the null distribution. Please double-check this and be more specific in the caption.
3. Figure 1F. Caption refers to the "mean gRNA effects". Does this "mean" refer to the average across all barcodes associated with the same gRNA or something else?
4. "All loci contributing to a background effect show epistasis with a perturbation (pairwise epistasis)". This sentence gave us pause because we thought this was a new statement. But we now believe that this sentence is meant as a generic truth (basically a restatement of the definition of epistasis) and serves as an introductory sentence for the discussion that follows. To reduce this confusion, it might be good to have a paragraph break here and/or change the language.
5. Figure 3D. Are these the actual fitness effects or the deviation values?
6. Figure 4a--d. We are very confused by how to interpret these figures. The y axis shows "mean fitness" but we were again confused by what the reference strain is with respect to which fitness is measured (see our major point 3). One standard and easily understandable measure of the fitness difference between alleles is the average selection coefficient of all segregants carrying the BY allele (plus the appropriate gRNA) relative to all segregants carrying the 3S allele (plus the same gRNA). But instead of reporting this simple measure, the authors appear to report the average fitness effect of each allele relative to some third strain. If this is the case, which alleles does this strain have at the focal locus? If it has one of the two alleles, then shouldn't either the green or the orange point be at zero in the control case? Then how come that both allelic effects are positive for the control case for Chr VII and Chr XIV Hubs? Please explain.

Furthermore, if fitness with respect to a third reference strain is reported, then one should be comparing the differences of these fitness values between each green and the corresponding orange point. But these plots make such comparisons difficult. The existing plots are more suitable for comparing values of points of the same color across gRNAs and the Control. If we understood this correctly, we suggest that the authors plot the difference of fitnesses of the two alleles instead of the fitness of each allele separately.
7. The discussion in the text associated with Figure 4 has a few confusing statements. For example, "and only individuals carrying the 3S allele are strongly affected by the gRNA". As far as we can tell from Figure 4b, the effects of the BY and 3S alleles (relative to some unknown reference strain) are similar in magnitude but different in sign. So, why do the authors say that the KDs of these genes in the BY background at the Chr X-Hub locus have no effect? The sentence "In the absence of gRNA activity, the Chr VII hub shows no significant effect on fitness" is similarly confusing. Here, the effects relative to the reference strain both seem to be significantly positive. Perhaps the authors refer to their difference. In this case, we agree, but then we again suggest to explicitly plot these differences.
8. We were surprised that the recent paper by Ang et al in Cell Genomics was not cited by the authors. It is a closely related study that uses a complementary high-throughput barcoding technique.

Reviewer #2 (Remarks to the Author):

A longstanding problem in genetics is understanding the influence of genetic background on the impact of genetic variants or perturbations. Apart from a basic scientific interest, finding rules governing this phenomenon is of great importance in the field of human genetics and the development of diagnostic and treatment solutions for many diseases. Yeast is an ideal model for these sort of studies as it allows for controlled crosses between haploid strains and has powerful genetic tools and fast growing times.

Hale and colleagues designed an interesting study by crossing two haploid genetic backgrounds and perturbed a large number of genes through CRISPRi. Thanks to a clever double barcoding strategy combined with an amplicon sequencing based assay they could record the impact of each perturbation across many different segregants (each of which has a unique genetic background). Consistent with earlier studies using different designs they observe how the impact of CRISPRi varies according to the background, and how it is difficult to find a single cause for these differences. They indicate a few patterns, such as the overall fitness of the segregant, the average impact of the perturbation, and a few modulating loci. The authors conclude that genetic background influences perturbation in a similar proportion of genes, as previously reported.

The design of this study is well thought out and convincing, and the conclusions, while not novel, are of interest because a new partially orthogonal assay is used. I want to raise some issues about the clarity of the text, the somewhat convoluted data processing pipeline, and the lack of detailed examples. I would consider the data processing comments as crucial to provide support to all the conclusions presented. If the results still hold after addressing data processing issues, then the other comments would need to be addressed to increase the value of the study and maybe even providing generalizable insights.

Data processing:

While the constructs used to barcode the segregants and perform the CRISPRi interference look solid, the assay itself and how the sequencing data was analyzed are hard to follow and have multiple points in which an ad-hoc approach has been taken with no justifications for the choices taken.

- the authors have serially passaged the three cultures and have apparently sequenced amplicons from all time points, but they obtain a single final deviation value. I could not understand if the time dynamics was part of the fitness estimation and if so how that was taken into account. The authors mention the use of a library called PyFitSeq; does that take into account the time component? More clarity is needed here, as having multiple time points looks like a sensible strategy to obtain strong estimates not affected by technical noise
- the authors refer to the use of custom python and R scripts, which should be shared in a public repository and their content detailed to some degree. I skimmed over the provided Google drive and noticed a heavy use of hardcoded paths and no simple README to indicate at least broadly what each file is supposed to do.
- I could not spot a sequence read archive ID for the raw sequencing data, which is crucial to aid people intending to replicate the data analysis pipeline
- there seems to be a large use of fuzzy pattern matching, sometimes with a high degree of "fuzziness" allowed (score of 90). it's unclear if this is bound to pool together very different barcodes, while an exact matching that relies on the fastq quality scores would be more straightforward. Did the authors test different thresholds, or have data showing how this non-orthodox processing of the data is appropriate and does not pool many barcodes together?
- "To minimize the effects of sequencing errors and random mutations on barcode counts, very similar barcodes were clustered together and treated as the same sequence." This seems to me like a bold claim, given that the error rate of Illumina and the mutation rate of yeast are known. How likely it is that these processes affect the recovered UMI/barcodes?
- The protocol used to assess the influence of PCR chimeras in barcode recovery seems sensible, although the estimated error rate seems quite high (~10%). This issue together with the many other ad-hoc corrections being made reduces the confidence in the resulting fitness/deviation estimates. Do the authors have any way to validate some of their results in an at least partially orthogonal way?
- the linkage mapping has been done on a rather small sample size, and it's unclear what the

power of this analysis is, and therefore how solid their conclusions on the hub loci they identify.

- the correlation between deviation and base fitness values and mean guide effects need to have confidence interval, as they might be driven by outliers, especially for the first correlation. It is also unclear if the correlation is drawn from all data points or just the average fitness for each segregant. The rather low p-value ($< 10^{-100}$) seems to indicate the former, but it really needs to be clarified.

Clarity: there are many places in the main text, figures, supplementary figures that make the flow hard to follow:

- How were the 169 segregants selected? If not randomly this needs to be specified in case the selection introduced some sort of bias

- Supplementary Figure 5/7, Supplemental data 5: the median should be reported as well, as I suspect it would be much lower than the mean and thus indicate how the distributions are skewed towards zero

- Supplementary Figure 6: the position of 3S is not indicated

- Figure 1D refers to ATC1, but in the text ATC2 is indicated, also for figure 1H. Which one is it? Typos like these reduce the confidence that no other errors in data processing are present.

- Figure 1E could be modified to overlay a stripplot on top of the bloxplot, given that there are not many data points to show.

- The permutations listed in the same paragraph could also be shown as an additional panel or another supplementary figure

- Supplementary figure 9B has another typo ("ATC1" instead of "ATC2")

- Why did the authors decide to focus on a single gRNA per gene? If that is because their effect is not correlated that would reduce the confidence in the conclusions substantially

- In Figure 2B the authors exclude rows/columns with more than 30% missing values: are those also excluded from the analysis? If not, how were missing values handled for the downstream analyses?

Lack of detailed examples: while the purpose of the study is broad, I feel that the authors could focus on a few examples to boost the confidence of the readers in their results. They do this by running functional enrichment analysis with CellMap, but choosing single examples for which the clearest possible mechanistic explanation for a deviation can be drawn would be far more useful. In particular, I found myself wanting for more details or a handful of examples in the following sections: "Properties of hub loci that interact with many genetic perturbations", sentence "This suggests that the causal genes underlying these two hubs may be impaired or nonfunctional in BY.". Can the authors actually look closely at the hubs? Later, "In summary, at both the Chr VII and XIV hubs, the fitness of individuals carrying the 3S alleles of these loci was substantially less impacted than the fitness of individuals carrying the BY alleles. This suggests that 3S carries alleles of these two hubs that may impart higher robustness to certain perturbations" seems to be a good place to look into some specific examples?

Other issues:

- can the authors provide more information about the issue that led to many gRNAs being depleted?

- supplementary figure 3 references supp. figure 11, but I'm sure the authors meant supp. figure 1

- the complete lack of reference points such as page and line numbers made reviewing this manuscript harder than it needed to be

To the Reviewers,

Thank you for your constructive feedback on our manuscript. We believe we have addressed all reviewer comments. Addressing your comments substantially improved the manuscript, in particular by enhancing precision and clarity. It also resulted in some changes to the results, although our overall findings remain qualitatively unchanged. Point-by-point responses to your comments follow. Also, we have provided versions of the revised manuscript with and without changes highlighted.

Sincerely,

Joseph Hale, Sasha Levy, and Ian Ehrenreich

REVIEWER COMMENTS

Reviewer #1 (Remarks to the Author):

Hale et al use a powerful double-barcoding system that they previously developed to systematically survey genetic interactions between natural genetic variation and CRISPRi gene knockdowns (KD). They measure the fitness effects of KDs of 1,721 genes in a panel of 169 yeast segregants and identify over 400 genes whose KDs display genetic-background dependence. Furthermore, they identify four "hub" loci where the natural genetic variation interacts with many KDs, and they characterize these interactions.

Although this paper does report anything particularly unexpected, it (along with other recent papers by the Ian Ehrenreich's and Hunter Fraser's groups) sets a new standard for the field. The barcoding and CRISPR-based genetic engineering techniques developed by the authors generate remarkable amounts of high-quality data which will undoubtedly serve as an important resource for the community. However, we identified several important methodological concerns which we think the authors should address before publication.

Thank you for these positive remarks.

1. We are concerned about the fact that the mean fitness of engineered strains in CON is significantly higher than zero (Figure 1H). Our understanding is that the introduction of the guide construct should have no fitness effect in the CON condition, at least in principle. Related to this, the strong correlation between fitness in CON and ATC (Figure 1d) is also surprising to us. The current explanation that this correlation arises "because only a subset of gRNAs have strong effects on fitness" does not make sense to us. To understand this better, we would have liked to see the correlation between two replicates in CON, if available. Again, we would expect no correlation between CON1 and CON2 replicates or between CON1 and ATC, if the introduction of the guide construct has no effect on fitness.

Can the authors offer some explanations for these somewhat strange observations? We can think of several potential explanations. (a) The authors measured fitness with respect to a low-fitness reference strain. (We did not understand what reference strain was used---see our major point 3 below.) If this is the case, it would seem appropriate to normalize this effect by setting the average fitness of all strains in CON to zero. (b) Systematic effects can come from noise in the reference strain frequency, e.g., if the reference barcodes are at low frequency. Noise in the reference introduces a systematic bias into the estimates of fitness effects of all mutations. (c) Guides are to some extent expressed even without induction. In this case, normalization with respect to the CON condition is

unnecessary and potentially obscures some interesting effects. (d) The introduction of the guide construct itself introduces a sizable fitness defect and the correlation in Figure 1d arises because the authors are measuring largely this effect which is common to CON and ATC rather than the effect of the induction. We think that some additional analyses can rule out some of these possibilities, particularly (d) which is the most concerning. At the very least, we feel that the authors need to directly address these issues in the text.

We would like to first address Reviewer #1's questions regarding how fitnesses were estimated, because our poor communication on this point resulted in a lot of downstream confusion. In our experiment, the fitness of a given strain in an assay was measured relative to the mean fitness of the population of all lineages in that assay, not relative to any specific strain. The approach we used is standard (e.g., see Levy et al. 2015. Nature; Schlecht et al. 2017. Nature Communications; Matsui et al. 2022. Nature Communications) and discussed in detail in Li, Salit, and Levy. Cell Systems. 2018. To improve clarity, we added a more explicit discussion of how the fitnesses were calculated and what they represent to the Results and Methods sections of our manuscript.

Reviewer #1 points out that the mean fitness in CON is nonzero, which we agree was a confusing way to present the data. This is a purely technical issue, as the fitnesses used in this study are all relative to the mean population fitness in a given assay, and would not necessarily be expected to be centered on zero. We have added a step to the data processing that adjusts all lineages such that the mean fitness of all lineages in the CON assay is exactly zero. This has no effect on the downstream analysis.

Lastly, Reviewer #1 points out that it is odd that fitnesses in ATC and CON are correlated, since gRNAs are only induced in the ATC condition. Indeed, in a CRISPR/CRISPRi experiment where the genotype of all cells in the population are identical and all gRNAs are efficacious, this would not be expected. However, in our experiment, there is a diversity of genotypes (and fitnesses) in each population and most gRNAs are not efficacious. The main source of variance in fitness in our data (in both CON and ATC) is variation in baseline fitness across segregants (Mullis et al. 2022. Genetics; Matsui et al. 2022. Nature Communications). Due to this variation in baseline fitness, fitnesses in CON and ATC should correlate well in the absence of any gRNA effects. Only when a gRNA is efficacious and has a fitness effect should we see a difference in fitness between the CON and ATC in our study. These efficacious gRNAs are a small minority in our study and prior work that uses the same gRNA set (Smith et al. 2016. Genome Biology; Smith et al. 2017. Molecular Systems Biology). These efficacious gRNAs are visualized in Figure 1G, Figure 1H, and Supplemental Figure 13. We have revised the Results section to clarify why the correlation between ATC and CON is expected under the conditions of this study.

2. Have the authors tested how stable their estimated fitness values are across time intervals? We are particularly concerned about such systematic variation because the CRISPRi perturbations are induced just before the assay. Several generations might be needed for the perturbed genes to be fully knocked down. If fitness estimates change over time, the authors could address this issue for example by considering only the fitness estimates derived from later time intervals.

We agree with Reviewer #1 that this is a potential source of noise in our data. We have tested the stability of the fitness estimates when different sets of time points are used and summarized the results in Supplemental Table 2. Given that our gRNA library targets a very wide variety of genes, we do expect that some gRNAs may have more rapid effects on fitness than others, and it is not possible to avoid this confounding factor entirely. However, the high correlations in Supplemental Table 2 imply that gRNA induction timing and the subset of time points used in fitness estimation did not impact our findings.

3. It is unclear to us what the authors mean by 'Fitness'. We are confused about it in two contexts. First, when the authors discuss the fitness of segregants, it appears that this fitness is measured with respect to a common reference strain or strains, but we could not figure out what these reference strains are. Please clarify this.

In our study, we followed the fitness estimation methods described in (Levy et al. 2015. Nature; Li, Salit, and Levy. Cell Systems. 2018) as closely as possible, but this was not well explained in the original manuscript. A more detailed explanation of the fitness estimation has been added to the Results and Methods. To directly address the question here, PyFitSeq calculates fitness values with respect to the mean fitness of the population at the initial time point in a time series. Because the composition of a population changes over time, mean fitness is not constant and, in the absence of any mutation, approaches the highest fitness lineage in the population over time. Lineages that start above the mean fitness are expected to rise in frequency until the mean fitness overtakes them, at which point they begin to decline in frequency at an accelerating rate. Lineages that start below the mean fitness are expected to decline in frequency at an accelerating rate. There is no specific reference strain used for fitness estimation.

Second, the authors state that there are 189 unique 'control' gRNAs which target intergenic regions. We initially thought that these were used as reference neutral mutations with respect to which fitness effects of KDs were estimated. But in the methods section on the 'quantification of gRNA effects' the authors state that they identify neutral gRNAs as all those with equal or higher fitness in ATC than control, and that these post-hoc identified 'neutral' gRNAs were used as for normalization. So, with respect to what is the fitness effect of each KD in each segregant estimated? Can the authors explain why they did not use the known putatively neutral 'control' gRNAs for normalization instead?

During preliminary analyses of the data and also during revision, we internally used several different normalization methods. All normalization strategies converged to similar results and had no qualitative impact on our findings. Normalization using control gRNAs that target intergenic and noncoding regions was one of these methods, but this process was not explained in the original manuscript, and the purpose of control gRNAs was not provided. In the revised manuscript, we make it more clear that we use the known control gRNAs for normalization of the three fitness assays (ATC1, ATC2, and CON). We revised the manuscript to more clearly explain the normalization of the three fitness assays using control gRNAs.

Regarding the question about how the fitness effect of each knockdown is estimated, the mixed effects linear models used to determine gRNA effect use the CON condition as a baseline or reference. To summarize, these models take all lineages carrying the same gRNA and compare their fitnesses across the ATC and CON conditions, while correcting out the fitness effects of each genotype. For example, a gRNA with a fitness effect of -0.5 would mean that our model estimates that induction of that gRNA causes fitness to drop by 0.5 in ATC compared to the same lineages in CON. Additional details on this process are given in the methods section 'Identification and quantification of gRNA effects', which has been revised in the updated manuscript.

MINOR POINTS

1. In the future, please add page and line numbers to make the lives of reviewers a bit easier.

Thank you for this feedback. Page and line numbers have both been added.

2. "We only considered a gRNA further if its fitness effect was at least three standard deviations below a null distribution inferred from the data". Do the authors mean "below the mean of the null

distribution”? It would be good to show this threshold in Figure 1F. There is a vertical line in Figure 1F, but we were not sure if this is actually the threshold discussed in the text because it seems to be less than 3 stds below the mean of the null distribution. Please double-check this and be more specific in the caption.

Thank you for this feedback, the section referring to Figure 1F has been fixed in the main text. The legend for this figure has been updated to be more descriptive as well. We have also double-checked to ensure the threshold is three standard deviations below the mean of the null distribution shown by the dashed line. The value used for analysis and the location of the vertical line on Figure 1F appear to be correct, so no other changes were made.

3. Figure 1F. Caption refers to the “mean gRNA effects”. Does this “mean” refer to the average across all barcodes associated with the same gRNA or something else?

That mean gRNA effect refers to the coefficient of the *gRNA* term in the mixed effects linear model, as described in the Methods section ‘Identification and quantification of gRNA effects’. We have added this point to both the figure legend and the Methods section.

4. “All loci contributing to a background effect show epistasis with a perturbation (pairwise epistasis)”. This sentence gave us pause because we thought this was a new statement. But we now believe that this sentence is meant as a generic truth (basically a restatement of the definition of epistasis) and serves as an introductory sentence for the discussion that follows. To reduce this confusion, it might be good to have a paragraph break here and/or change the language.

Thank you for the feedback. We inserted a paragraph break as suggested.

5. Figure 3D. Are these the actual fitness effects or the deviation values?

The histogram in Figure 3D shows the effect of each locus on deviation value. Specifically, these values were obtained from the linear model *deviations* ~ *locus*, taking the coefficient of the *locus* term. This is described in more detail in the Methods section ‘Linkage mapping’. This value is roughly equivalent to the difference between the mean deviation value of segregants carrying the 3S allele and the mean deviation value of segregants carrying the BY allele. Additional clarification has been added to the legend of Figure 3.

6. Figure 4a--d. We are very confused by how to interpret these figures. The y axis shows “mean fitness” but we were again confused by what the reference strain is with respect to which fitness is measured (see our major point 3). One standard and easily understandable measure of the fitness difference between alleles is the average selection coefficient of all segregants carrying the BY allele (plus the appropriate gRNA) relative to all segregants carrying the 3S allele (plus the same gRNA). But instead of reporting this simple measure, the authors appear to report the average fitness effect of each allele relative to some third strain. If this is the case, which alleles does this strain have at the focal locus? If it has one of the two alleles, then shouldn’t either the green or the orange point be at zero in the control case? Then how come that both allelic effects are positive for the control case for Chr VII and Chr XIV Hubs? Please explain.

Furthermore, if fitness with respect to a third reference strain is reported, then one should be comparing the differences of these fitness values between each green and the corresponding orange point. But these plots make such comparisons difficult. The existing plots are more suitable for comparing values of points of the same color across gRNAs and the Control. If we understood this

correctly, we suggest that the authors plot the difference of fitnesses of the two alleles instead of the fitness of each allele separately.

We agree with Reviewer #1 that Figures 4A-D were confusing, and have substantially modified these plots. We greatly appreciate the feedback about this figure, and believe with Reviewer #1's help we have arrived at a more intuitive way to show the data that should address concerns. The upper panel of each plot now shows mean fitness for different genotype classes at each gRNA that interacts with a hub. The corresponding deviation values are shown separately, on the lower panel of each plot. This makes the effect of each interacting gRNA directly visible, and shows how the induction of each gRNA alters the effects of the hub. We attempted Reviewer #1's suggestion to only plot the difference in fitnesses between the two genotype classes, but in our opinion this does not illustrate the effects of each hub and each gRNA as effectively. Figures 4A-D, the results section 'Properties of hub loci that interact with many genetic perturbations', and Supplemental Figure 16 have all been updated. As an additional point of clarification, the fitness measurements used in these plots do not involve any specific reference strain, which is explained in more detail in the response to point #3 above.

Also, while revising Figure 4, we noticed an error with the original version of this figure, which has been corrected in the updated manuscript. In the previous version of Figure 4, the panel with the control lineages (gray background) showed mean fitness, and the panel with the interacting gRNAs (white background) showed deviation values. The two panels were inadvertently combined, as if they shared the same measurement on the y-axis. While all values shown were accurate, this led to an incorrect interpretation of the effects of the hubs in the corresponding section of the results. In the current version of Figure 4, mean fitness is correctly shown for both the control lineages and the interacting gRNAs, and this section of the results has been rewritten to match. We thank Reviewer #1 (and #2) for bringing this mistake to our attention. We hope that the revised figure and text addresses the questions posed here and is clearer to understand.

7. The discussion in the text associated with Figure 4 has a few confusing statements. For example, "and only individuals carrying the 3S allele are strongly affected by the gRNA". As far as we can tell from Figure 4b, the effects of the BY and 3S alleles (relative to some unknown reference strain) are similar in magnitude but different in sign. So, why do the authors say that the KDs of these genes in the BY background at the Chr X-Hub locus have no effect? The sentence "In the absence of gRNA activity, the Chr VII hub shows no significant effect on fitness" is similarly confusing. Here, the effects relative to the reference strain both seem to be significantly positive. Perhaps the authors refer to their difference. In this case, we agree, but then we again suggest to explicitly plot these differences.

We agree with Reviewer #1 that the language in this section was too confusing and vague. As noted above, this section of the results has been entirely rewritten as a result of the changes to Figure 4. We have taken this feedback into account during the rewrite, and have explained in more detail how we arrived at the interpretation of each hub's effects. As discussed above, we tried explicitly plotting differences between genotype classes, but found that this made the effects of the interacting gRNAs difficult to interpret. We hope that what we now include as Figure 4 is a more effective visualization. These revisions also led us to display the CellMap results as a separate figure (Figure 5), with each hub plotted on its own.

8. We were surprised that the recent paper by Ang et al in Cell Genomics was not cited by the authors. It is a closely related study that uses a complementary high-throughput barcoding technique.

Thank you for noticing this. We have added the citation.

Reviewer #2 (Remarks to the Author):

A longstanding problem in genetics is understanding the influence of genetic background on the impact of genetic variants or perturbations. Apart from a basic scientific interest, finding rules governing this phenomenon is of great importance in the field of human genetics and the development of diagnostic and treatment solutions for many diseases. Yeast is an ideal model for these sort of studies as it allows for controlled crosses between haploid strains and has powerful genetic tools and fast growing times.

Hale and colleagues designed an interesting study by crossing two haploid genetic backgrounds and perturbed a large number of genes through CRISPRi. Thanks to a clever double barcoding strategy combined with an amplicon sequencing based assay they could record the impact of each perturbation across many different segregants (each of which has a unique genetic background). Consistent with earlier studies using different designs they observe how the impact of CRISPRi varies according to the background, and how it is difficult to find a single cause for these differences. They indicate a few patterns, such as the overall fitness of the segregant, the average impact of the perturbation, and a few modulating loci. The authors conclude that genetic background influences perturbation in a similar proportion of genes, as previously reported.

The design of this study is well thought out and convincing, and the conclusions, while not novel, are of interest because a new partially orthogonal assay is used. I want to raise some issues about the clarity of the text, the somewhat convoluted data processing pipeline, and the lack of detailed examples. I would consider the data processing comments as crucial to provide support to all the conclusions presented. If the results still hold after addressing data processing issues, then the other comments would need to be addressed to increase the value of the study and maybe even providing generalizable insights.

Thank you for this feedback. We agree that several sections of the text were unclear, and have reviewed the data processing pipeline used in this study. Point-by-point responses are provided below.

Data processing:

While the constructs used to barcode the segregants and perform the CRISPRi interference look solid, the assay itself and how the sequencing data was analyzed are hard to follow and have multiple points in which an ad-hoc approach has been taken with no justifications for the choices taken.

We agree that the data processing was not sufficiently explained in the original manuscript. The experimental design of the fitness assay and fitness estimation was based on prior publications (e.g., see Levy et al. 2015. Nature; Schlecht et al. 2017. Nature Communications; Li, Salit, and Levy. Cell Systems. 2018; Matsui et al. 2022. Nature Communications). However, many of these steps were not explained or justified in the original manuscript. Based on multiple reviewer comments, we believe this lack of detail has made several standard steps appear to be ad-hoc; thus, significant revisions to the results and methods sections were made to provide justifications for most major steps in data processing. Also, to make our analysis pipeline more interpretable, a large number of comments have been added to all sections of code used for analysis.

- the authors have serially passaged the three cultures and have apparently sequenced amplicons from all time points, but they obtain a single final deviation value. I could not understand if the time dynamics was part of the fitness estimation and if so how that was taken into account. The authors mention the use of a library called PyFitSeq; does that take into account the time component? More clarity is needed here, as having multiple time points looks like a sensible strategy to obtain strong estimates not affected by technical noise

PyFitSeq does take multiple time points into account, even though a single fitness estimate is produced for each lineage. As mentioned in a response to Reviewer #1, multiple time points are used to estimate the fitness of each segregant relative to the mean population fitness, which varies over time. We agree that the methods of fitness estimation used in this study were not well explained, and a more detailed explanation of this process has been added to the Results and Methods.

- the authors refer to the use of custom python and R scripts, which should be shared in a public repository and their content detailed to some degree. I skimmed over the provided Google drive and noticed a heavy use of hardcoded paths and no simple README to indicate at least broadly what each file is supposed to do.

Thank you for the feedback on this matter. The custom R and python scripts have been edited to reduce hardcoded paths where possible, and a README describing the function of each file has also been added. Additional comments and descriptions have also been added to most scripts. All scripts have also been uploaded to a Figshare repository that will become public when the manuscript is published and should presently be available to Reviewer #2 via a link supplied by Nature Communications.

- I could not spot a sequence read archive ID for the raw sequencing data, which is crucial to aid people intending to replicate the data analysis pipeline

The Sequence Read Archive ID PRJNA986287 and associated information has been added to the manuscript.

- there seems to be a large use of fuzzy pattern matching, sometimes with a high degree of "fuzzyness" allowed (score of 90). it's unclear if this is bound to pool together very different barcodes, while an exact matching that relies on the fastq quality scores would be more straightforward. Did the authors test different thresholds, or have data showing how this non-orthodox processing of the data is appropriate and does not pool many barcodes together?

We agree that this section of data processing was poorly explained. In our experiment, we already knew the true barcodes that were present in each assay. Fuzzy matching was primarily used to correct discrepancies between Bartender clusters and these known barcodes. Clustering reads is a standard step in processing barcode sequencing data, as it accounts for sequencing errors and mutations relative to known barcodes (Zhao et al. 2018. Bioinformatics). This is discussed in more detail in the next response. We found that Bartender would sometimes collapse reads into clusters that did not perfectly match known barcodes. To address this problem, we used fuzzy matching with a score threshold of 90. This score is not a degree of fuzziness allowed, but rather a degree of similarity required, with a score of 100 indicating two perfectly identical strings. We believe our required similarity score of 90 is conservative, as this should correspond to a roughly 2 nucleotide difference. In our data set, the average Hamming distance between two barcodes is >10. We also note that over 95% of reads clustered this way are associated with a perfectly matched barcode (score = 100), indicating that this is a relatively minor step of data processing. However, to ensure that we did not pool dissimilar reads together, we examined the similarity scores between these Bartender clusters and the two closest matches among known barcodes. The average score of the second-best match was substantially worse than the best match (84.7 versus 96.7), suggesting that over-clustering is not a concern. A more detailed explanation of this fuzzy matching step has been added to the Methods.

- "To minimize the effects of sequencing errors and random mutations on barcode counts, very similar barcodes were clustered together and treated as the same sequence." This seems to me like a bold

claim, given that the error rate of Illumina and the mutation rate of yeast are known. How likely it is that these processes affect the recovered UMI/barcodes?

Clustering of very similar barcodes via software like Bartender is a standard process when analyzing barcode sequencing data (Levy et al. 2015. Nature; Schlecht et al. 2017. Nature Communications; Matsui et al. 2022. Nature Communications). This process is primarily meant to reduce bias introduced by the uneven distribution of sequencing and PCR errors across the 20-nucleotide barcodes, some of which are more error-prone than others. If our analysis of the sequencing data only considered reads that contained a perfect match to a known barcode, then any barcodes with error-prone sequences could potentially appear at an artificially-lowered frequency. This was a particular concern for our study, as barcodes consistently dropping to lower frequency than expected is how we detected gRNAs with effects. We have double-checked the publications cited above that also use Bartender for their data processing, and given that our methods are similar, we have not altered this step of the analysis.

The wording of this question by Reviewer #2 also implies that UMIs and barcodes were treated similarly in our data set. As an additional point of clarification, we use 'UMI' to refer to the 8-nucleotide sequences that mark unique amplicons in our sequencing data. 'Barcode' refers to the roughly 20-nucleotide sequences that represent either a specific genotype or a specific integrated gRNA. Barcodes and UMIs do not undergo any of the same steps of analysis, and only barcodes ever undergo the clustering process. Clustering of UMIs is never performed under any circumstance. This distinction may be unintuitive, especially because the two terms are occasionally used interchangeably in the literature, so we have added a short clarification of this to the methods.

- The protocol used to assess the influence of PCR chimeras in barcode recovery seems sensible, although the estimated error rate seems quite high (~10%). This issue together with the many other ad-hoc corrections being made reduces the confidence in the resulting fitness/deviation estimates. Do the authors have any way to validate some of their results in an at least partially orthogonal way?

The high rate of PCR chimeras is a technical issue we were concerned with as well. In our study, the two random barcodes present in each amplicon are separated by a fixed region of roughly 100 bases. In addition to our own experiences, previous studies have indicated that this amplicon design can lead to high rates of PCR chimeras, with some reporting values as high as 30% (Omelina et al. 2019. BMC Genomics). The exact PCR chimera rate is difficult to estimate without a specialized experiment, but we do not observe any indications that PCR chimeras are affecting our fitness measurements. If PCR chimeras were significantly impacting the fitness estimates of most lineages, we would expect to see very poor correlation between the two replicate ATC flasks as a result of random noise. However, the two ATC flasks have highly correlated fitness estimates and deviation values generally show high heritability in our data (Figures 1C and G), and have correlations similar to those observed in previous publications (Matsui et al. 2022. Nature Communications). Thus, we do not expect PCR chimeras to have a significant impact on the results presented.

As a point of clarification, the value given in the methods (<10%) is the average number of chimeric double barcodes in the tested segregants, with the exact value being 7.26%. In other words, this is the number of reads with an invalid combination of two barcodes. The exact chimera rate is considerably more difficult to estimate, as not all chimeras result in an invalid combination of barcodes in our experiment, and we did not attempt to directly calculate this value. This sentence has been revised in the methods section for greater clarity.

We point to two sources of orthogonal validation from independent projects in different labs. First, there was substantial overlap between gRNAs with effects in our study and gRNAs identified with effects in a prior study using the same library of genetic perturbations (Smith et al. 2016. Genome

Biology). We note our study involved many more strains, providing more power to detect gRNAs with effects. Second, the group of Michael Desai used transposon mutagenesis to introduce genetic perturbations into a variety of genotypes in a similar manner as our study. They found loci similar to our hubs, which they referred to as multi-hit loci (Johnson et al. 2019. Science). Specifically, the hubs on chromosome XIII and XIV overlap multi-hit loci identified in Johnson et. al, in addition to two potential hubs that fell slightly below our significance threshold (Supplemental Figure 16). We now point to these corroborating studies in the Results section. We would also stress the high reproducibility of our experiment across replicate assays and the high heritability values we obtained for deviation values within assays as additional means to validate our results.

- the linkage mapping has been done on a rather small sample size, and it's unclear what the power of this analysis is, and therefore how solid their conclusions on the hub loci they identify.

Assuming that Reviewer #2 is suggesting the hubs may be false positives, this is a possibility we were concerned with as well. However, several results have led us to believe it is unlikely that the hubs are false positives. First, we detect the same hubs when we analyze ATC1 and ATC2. This information was not present in the original manuscript, and has been added as Supplemental Figure 15. Second, as mentioned above, several of our detected hubs directly overlap multi-hit loci found in the Johnson et al. 2019. Science study. Third, the hubs represent loci detected for many different gRNAs, providing another level of replication.

Regarding the issue of power, we agree that low statistical power is a limitation of our study. There are very likely more loci that contribute to the background effects identified in this study. However, statistical power regards the chance of detecting an effect that is actually present, and should not affect false positive rate. Even if more loci are found in future studies with higher power, the fact that the large-effect loci identified here cluster into hubs more often than expected by chance should remain true. The failure to detect these other potential loci can only be addressed by performing the study with a much larger number of segregants. While such a study was not possible when we conducted our initial work, we currently plan to perform such an experiment using the gRNAs for which we detected background effects.

- the correlation between deviation and base fitness values and mean guide effects need to have confidence interval, as they might be driven by outliers, especially for the first correlation. It is also unclear if the correlation is drawn from all data points or just the average fitness for each segregant. The rather low p-value ($< 10^{-100}$) seems to indicate the former, but it really needs to be clarified.

Thank you for this feedback, we agree that a confidence interval is necessary. Confidence intervals as determined by bootstrapping have been added to the legends of Figures 2C and 2D.

Clarity: there are many places in the main text, figures, supplementary figures that make the flow hard to follow:

- How were the 169 segregants selected? If not randomly this needs to be specified in case the selection introduced some sort of bias

The *MATa* segregants used in this study were randomly selected from different tetrads. However, this is only mentioned very briefly in the introduction. This has also been added to the methods section 'Transformation of haploid strains with the CRISPRi library'.

- Supplementary Figure 5/7, Supplemental data 5: the median should be reported as well, as I suspect it would be much lower than the mean and thus indicate how the distributions are skewed towards zero

The median is now reported in the legends of Supplemental Figures 5 and 7.

- Supplementary Figure 6: the position of 3S is not indicated

The transformation efficiency of the 3S parent has been added to this figure.

- Figure 1D refers to ATC1, but in the text ATC2 is indicated, also for figure 1H. Which one is it? Typos like these reduce the confidence that no other errors in data processing are present.

Thank you for this feedback, this typo has been corrected. This error was introduced when we decided to switch the names of ATC1 and ATC2 while writing the manuscript. We have confirmed that it did not affect any data processing or analysis.

- Figure 1E could be modified to overlay a stripplot on top of the bloxplot, given that there are not many data points to show.

Thank you for this feedback, a stripplot has been overlaid onto Figure 1E.

- The permutations listed in the same paragraph could also be shown as an additional panel or another supplementary figure

Thank you for this feedback, additional information about the permutations has been added to Supplemental Figure 12.

- Supplementary figure 9B has another typo ("ATC1" instead of "ATC2")

Thank you for this feedback, this typo has been corrected.

- Why did the authors decide to focus on a single gRNA per gene? If that is because their effect is not correlated that would reduce the confidence in the conclusions substantially

This is an excellent question. The reason we did this is not due to uncorrelated effects of gRNAs. In fact, the effects of gRNAs targeting the same genes are usually correlated, as shown in Supplemental Figure 12. However, we tried to focus our work towards the gRNAs with strongest effects on target genes, as inferred from fitness. Efficacious gRNAs targeting the same gene do not always have the same effect because some gRNAs are worse at blocking transcription, reducing how much knockdown occurs. This is expected with CRISPRi, as gRNAs are often positioned in a variety of orientations and distances from a transcription start site. This can cause gRNAs to vary considerably in their effects on transcription and is a reason why many of our gRNAs are not efficacious.

Also, we did initially perform the analysis using all gRNAs, regardless of how many targeted the same gene. This produced qualitatively the same results and conclusions shown in the current manuscript. However, we found it was easier for readers to understand the results when we included only a single gRNA per gene, especially since this removes only a small amount of data (<20% of mapped loci). We also found that certain analyses were more straightforward. For example, defining hubs was more intuitive when only a single gRNA was included per targeted gene, as it was not clear how to account for some genes having more than one gRNA in setting a threshold for calling hubs. To ensure this did not impact our results, we selected an alternate gRNA with a mapped locus for the 53 genes for which one was available (out of 460 genes targeted by efficacious gRNAs). All four hubs are still detected under these conditions.

- In Figure 2B the authors exclude rows/columns with more than 30% missing values: are those also excluded from the analysis? If not, how were missing values handled for the downstream analyses?

This threshold was due to an issue with visualizing the heatmap. Due to the scale of this experiment, most segregants lack at least some gRNAs (in other words, most gRNAs are not present in all segregants). Such missing data can make it hard to read the heatmap. Thus, we used a missing data threshold solely for the purpose of visualization in Figure 2B. That is the only place it comes into play. All analyses performed in the paper use all available data.

Lack of detailed examples: while the purpose of the study is broad, I feel that the authors could focus on a few examples to boost the confidence of the readers in their results. They do this by running functional enrichment analysis with CellMap, but choosing single examples for which the clearest possible mechanistic explanation for a deviation can be drawn would be far more useful. In particular, I found myself wanting for more details or a handful of examples in the following sections: "Properties of hub loci that interact with many genetic perturbations", sentence "This suggests that the causal genes underlying these two hubs may be impaired or nonfunctional in BY.". Can the authors actually look closely at the hubs? Later, "In summary, at both the Chr VII and XIV hubs, the fitness of individuals carrying the 3S alleles of these loci was substantially less impacted than the fitness of individuals carrying the BY alleles. This suggests that 3S carries alleles of these two hubs that may impart higher robustness to certain perturbations" seems to be a good place to look into some specific examples?

As we discussed above in the response to Reviewer #1, the original version of Figure 4 contained an error. While all values shown were accurate, the panels indicating the fitness of the control lineages (gray background) were inadvertently plotted on the same y-axis as the deviation values for interacting gRNAs (white background). The results were written with an incorrect interpretation of Figure 4, including the quotes Reviewer #2 references above. We thank Reviewer #2 (and #1) for bringing this mistake to our attention, and we have significantly changed Figure 4 and its associated text to more clearly show how gRNAs interact with a hub locus. This new presentation directly shows how each gRNA has a large fitness effect regardless of the allele at the hub locus. However, the magnitude of this fitness effect depends on the hub locus allele. We hope that this new presentation is simpler to understand and boosts the confidence of the reviewer in the results.

We agree with the reviewer that specific mechanistic examples of gRNA-hub interactions would have been preferable, and we thoroughly investigated whether any connections could be made. However, the limited number of recombinations that occur within a single cross (as performed here) resulted in a poor mapping resolution (loci were mapped at a resolution of >20 genes). This poor resolution prevented us from connecting a specific gene in each hub to the genes targeted by gRNAs with any confidence. We attempted to use existing genetic and physical interaction data (from BIOGRID and CellMap) to help us make these connections. However, we could not find any strong candidates and thus did not discuss this effort in the revision.

Other issues:

- can the authors provide more information about the issue that led to many gRNAs being depleted?

We assume that Reviewer #2 is referring to the depletion of nonessential gRNAs referenced in the methods section 'Generation of the CRISPRi library'. We had originally intended for the gRNA plasmid library to target all genes in *S. cerevisiae*, with the plasmid libraries targeting essential and nonessential genes being synthesized separately. The essential library was generated first, and we identified an issue where the large number of nucleotides between the gRNA and its corresponding

barcode made sequencing the two in a single Illumina read very difficult, because amplicons were too long. This required using Oxford Nanopore sequencing to verify the essential gRNA library, which is not ideal because of the large size of these barcoded gRNA libraries (we had >20 different barcodes per gRNA) and the relatively high error rate of Oxford Nanopore sequencing. With this in mind, the nonessential library used a slightly different plasmid structure which reduced this distance significantly. This had the side effect of placing the promoter for URA3 (the selectable marker used when the plasmids were integrated in yeast) much closer to the gRNA's terminator sequence. After a pilot experiment where we pooled both plasmid libraries together, integrated them into yeast strains, grew them in selective SC-URA media, and sequenced the resulting double barcodes, we found that most gRNAs from the nonessential library were at significantly lower frequency than expected. Additional tests indicated that strains carrying plasmids from the nonessential library grew noticeably slower on SC-URA plates when compared to strains carrying plasmids from the essential plasmid library. We believe the most likely explanation to be reduced URA3 expression in the nonessential library, as a result of the close proximity between its promoter and the gRNA's terminator. Given that roughly 10.5% of our data consisted of nonessential gRNAs, and it was not feasible to remake the entire plasmid library and repeat all strain generation, we chose to proceed with the experiment and exclude the nonessential gRNAs from analysis.

- supplementary figure 3 references supp. figure 11, but I'm sure the authors meant supp. figure 1

Thank you for this feedback, this typo has been corrected.

- the complete lack of reference points such as page and line numbers made reviewing this manuscript harder than it needed to be

We have added page and line numbers to the revision.

REVIEWER COMMENTS

Reviewer #1 (Remarks to the Author):

We appreciate the time and effort the authors have put into addressing our comments. Most of our issues have been resolved, and the clarity of the manuscript has substantially improved. There is however one issue that remains a concern, namely the possible variability of the effects of perturbations across time.

The new Supplementary Table 2 is helpful, but it does not address our concern. There are two problems with the approach taken by the authors. First, high Pearson correlation does not imply that perturbations have the same effect when measured over different time intervals. For example, the Pearson correlation coefficient between X and $Y = 2X$ is 1, but of course $Y \neq X$. The second problem is that estimates in all but one comparison in this table are non-independent because the effects are estimated from overlapping sets of time intervals. As a result, the correlation coefficients in this table are inflated. The only comparison of independent time intervals (T0-T1-T2 vs T2-T3-T5-T7) has by far the lowest correlation. A lower correlation is expected when fewer time intervals are used for the estimation. However, this is probably not the main reason for low correlation, since other 2 versus 3 time-interval comparisons (where intervals are overlapping) have much higher correlations. Thus, the low correlation in the T0-T1-T2 vs T2-T3-T5-T7 comparison suggests time dependence that we are concerned about.

To address this concern, we would kindly ask the authors to do two things. 1) Provide a plot or plots of correlations between the effect estimates obtained in different non-overlapping time intervals, particularly "early" versus "late" estimates, i.e., a supp figure analogous to 1c but with early versus late fitness estimate in ATC. 2) If the authors find evidence of time-dependence of perturbation effects, we ask that they acknowledge this limitation of their approach and discuss how it affects their main results. In particular, we suspect that "late" fitness estimates should be more robust because most knock-downs have achieved their equilibrium values, and so the results obtained from these later time points are probably more reproducible and biologically meaningful.

Our opinion is that the existence of time dependence does not necessarily undermine the results of this manuscript or the utility of the method more generally. However, we think that the main value of this paper is methodological, in that it demonstrates the power of detecting genetic interactions between many individual perturbations and segregating genetic variants at scale. Thus, we think it is important and useful for potential future users that the authors acknowledge and explicitly discuss the limitations of their approach.

MINOR COMMENTS:

P. 9. "We only considered a gRNA further if its fitness effect was...". Do the authors mean the average fitness effect of gRNA across all segregants? It would be good to clarify this.

P. 10. "as most efficacious gRNAs (~54%) have indistinguishable effects across segregants". We think the authors mean "the majority of efficacious gRNAs".

Figure 2C. This result is consistent with the results by Johnson et al (Ref. 6). It would be helpful for the reader to draw this connection explicitly.

Figure 2D. This correlation could be explained by the fact that perturbations with stronger effects (i.e., lower fitness) are present at lower abundances in the pool, which would naturally make their estimates noisier. It would be good to mention this simple technical explanation.

Figure 3B. We found it curious that the majority of gRNAs have zero narrow-sense heritability (peak at 1). We think it might be worth pointing out this little fact in the Results section and discuss what it means in the Discussion.

P. 12, last paragraph. "The hubs on Chr X and Chr XIII each overlapped one of these fitness loci, implying that the same locus can affect both baseline segregant fitness and the effects of genetic

perturbations." Given the correlation between the baseline segregant fitness and the effects of perturbations (Figure 3C), we wonder whether the effects of Chr X and Chr XIII hubs on perturbations are mediated entirely through baseline segregant fitness or go above and beyond segregant fitness.

Figure 4. How are gRNAs ordered on the x axis?

P. 34. 83 control gRNAs are mentioned but not described in the text. What are these?

P. 34-35. The normalization procedure could be simplified. The authors essentially measure relative fitness with respect to the 83 control gRNAs, except the fitness of these control gRNAs now takes some non-zero (but arbitrary) value. It would be more natural and simpler to set the fitness of these control gRNAs to zero in both CON and each of ATC conditions.

Reviewer #2 (Remarks to the Author):

I would like to thank the authors for their very detailed response, which in my opinion has helped improve the manuscript and clarify several points. This will in turn help readers better understand the strengths and limitations of this study. I can't help feeling that the initial submission could have been more carefully checked for errors, as the change in Figure 4 very well exemplifies. I would also like to thank the authors for sharing their analysis code and having added the bare minimum amount of guidance for anyone wishing to review it and perhaps reuse it (although I could see that virtually all paths are still absolute and therefore not going to work on someone else's system).

February 16, 2024

To the Reviewers:

Thank you for the feedback on our revision. We addressed the remaining comments from Reviewer 1 below. With this resubmission, we included two versions of the manuscript, one with changes highlighted and the other without.

Sincerely,

Joseph Hale, Sasha Levy, and Ian Ehrenreich

REVIEWER COMMENTS

Reviewer #1 (Remarks to the Author):

We appreciate the time and effort the authors have put into addressing our comments. Most of our issues have been resolved, and the clarity of the manuscript has substantially improved. There is however one issue that remains a concern, namely the possible variability of the effects of perturbations across time.

The new Supplementary Table 2 is helpful, but it does not address our concern. There are two problems with the approach taken by the authors. First, high Pearson correlation does not imply that perturbations have the same effect when measured over different time intervals. For example, the Pearson correlation coefficient between X and $Y = 2X$ is 1, but of course $Y \neq X$. The second problem is that estimates in all but one comparison in this table are non-independent because the effects are estimated from overlapping sets of time intervals. As a result, the correlation coefficients in this table are inflated. The only comparison of independent time intervals (T0-T1-T2 vs T2-T3-T5-T7) has by far the lowest correlation. A lower correlation is expected when fewer time intervals are used for the estimation. However, this is probably not the main reason for low correlation, since other 2 versus 3 time-interval comparisons (where intervals are overlapping) have much higher correlations. Thus, the low correlation in the T0-T1-T2 vs T2-T3-T5-T7 comparison suggests time dependence that we are concerned about.

To address this concern, we would kindly ask the authors to do two things. 1) Provide a plot or plots of correlations between the effect estimates obtained in different non-overlapping time intervals, particularly "early" versus "late" estimates, i.e., a supp figure analogous to 1c but with early versus late fitness estimate in ATC. 2) If the authors find evidence of time-dependence of perturbation effects, we ask that they acknowledge this limitation of their approach and discuss how it affects their main results. In particular, we suspect that "late" fitness estimates should be more robust because most knock-downs have achieved their equilibrium values, and so the results obtained from these later time points are probably more reproducible and biologically meaningful.

Our opinion is that the existence of time dependence does not necessarily undermine the results of this manuscript or the utility of the method more generally. However, we think that the main value of this paper is methodological, in that it demonstrates the power of detecting genetic interactions between many individual perturbations and segregating genetic variants at scale. Thus, we think it is important and useful for potential future users that the authors acknowledge and explicitly discuss the limitations of their approach.

Thank you for this detailed feedback. We added comparisons of mean gRNA effect and fitness estimates between T0-T1-T2 ('early'), T2-T3-T5 ('late'), and T0-T1-T2-T3-T5 ('all') to the paper as Supplemental Figures 14 and 15. We could not analyze completely non-overlapping time series because T7 was not sequenced in all assays and a minimum of three time points are needed for fitness estimation. Text describing the time dependence of fitness estimation and rationale for why we used the time points we did has also been added to the Results section on page 9. Text describing limitations associated with time dependence and its impact on our results has been added to the Discussion section on page 15.

Reviewer #1 hypothesized that mean gRNA effect estimates from late time points would have greater reproducibility and magnitudes. However, mean gRNA effect estimates were in fact highly correlated between replicate assays regardless of the time points chosen (Supplemental Figure 15A-C). Consistent with Reviewer #1's hypothesis, we did find that late time points produce higher mean gRNA effect magnitudes than either early time points or all time points (Supplemental Figure 14A-C).

We also found that late time points caused issues with the reproducibility of fitness estimates, which is a concern for our study focused on the genotype-specific effects of perturbations. Fitness estimates of low frequency lineages are known to be less precise than those of more abundant lineages (Levy et al. 2015. Science; Li et al. 2018. Cell Syst; Matsui et al. 2022. Nat Comms). In our study, lineages with efficacious gRNAs exhibit the lowest frequencies at late time points due to the large fitness effects of gRNAs. When only late time points are used in fitness estimation, these lineages with efficacious gRNAs show low reproducibility in their fitness estimates (Supplemental Figure 15D-E). However, as Reviewer #1 notes, a confounding factor is that fitness estimates based on fewer time points will also show decreased reproducibility. Regardless of cause, less reproducible fitness estimates would have negatively impacted our analyses.

Importantly, mean gRNA effect magnitudes were highly correlated between all and late time points (Spearman's $\rho = 0.804$). Because our analysis focused on the relative effects of gRNAs across segregants, consistent underestimation of mean gRNA effect magnitude is unlikely to qualitatively impact our findings. Further, all and late time points largely resulted in detection of the same efficacious gRNAs. 88% (1472 of 1675) of the gRNAs that were identified as efficacious using late time points were also detected using all time points.

In summary, to balance our needs for accurate mean gRNA effect estimates and reproducible fitness estimates, we used all time points in our analyses.

MINOR COMMENTS:

P. 9. "We only considered a gRNA further if its fitness effect was...". Do the authors mean the average fitness effect of gRNA across all segregants? It would be good to clarify this.

Yes, that is correct. We have modified the text to consistently use the term 'mean effect' when we refer to the average fitness effect of a gRNA across all segregants.

P. 10. "as most efficacious gRNAs (~54%) have indistinguishable effects across segregants". We think the authors mean "the majority of efficacious gRNAs".

We have made the suggested modification.

Figure 2C. This result is consistent with the results by Johnson et al (Ref. 6). It would be helpful for the reader to draw this connection explicitly.

Thank you for this feedback, this connection has been added on page 11.

Figure 2D. This correlation could be explained by the fact that perturbations with stronger effects (i.e., lower fitness) are present at lower abundances in the pool, which would naturally make their estimates noisier. It would be good to mention this simple technical explanation.

We agree that this could be a contributor to the relationship shown in Figure 2D. We have added a sentence to the Results section, on page 11.

Figure 3B. We found it curious that the majority of gRNAs have zero narrow-sense heritability (peak at 1). We think it might be worth pointing out this little fact in the Results section and discuss what it means in the Discussion.

We agree with Reviewer #1 that this finding is curious, but would point out that these 35 gRNAs constitute only a small minority (7.6%) of all gRNAs with background effects. A potential biological explanation for their zero narrow-sense heritability could be that the effect of a gRNA depends on higher-order epistasis between multiple loci and a gRNA, similar to some of our previous work (e.g., Taylor and Ehrenreich. 2014. PLOS Genetics; Taylor and Ehrenreich. 2015. PLOS Genetics; Lee et al. 2016. PLOS Genetics; Taylor et al. 2016. Nat Comms; Lee et al. 2019. Genetics). It is also possible that these results represent a technical artifact with estimating narrow-sense heritability using the sommer package in R (Covarrubias-Pazaran. 2016. PLOS ONE). We suspect that there is considerable noise in our narrow-sense heritability estimates based on the range of values observed. To respond to Reviewer #1's comment, we added sentences to the Results (page 12) and Discussion (page 16).

P. 12, last paragraph. "The hubs on Chr X and Chr XIII each overlapped one of these fitness loci, implying that the same locus can affect both baseline segregant fitness and the effects of genetic perturbations." Given the correlation between the baseline segregant fitness and the effects of perturbations (Figure 3C), we wonder whether the effects of Chr X and Chr XIII hubs on perturbations are mediated entirely through baseline segregant fitness or go above and beyond segregant fitness.

We cannot conclusively answer this question with the current data. Because of our poor mapping resolution, it remains possible that the Chr X and XIII hubs each harbor distinct, linked variants that impact baseline fitness and response to genetic perturbations. However, if we assume that the same variant at each locus impacts both baseline fitness and response to genetic perturbations, our study could suggest the interactions between the hubs and perturbations are not entirely mediated through baseline segregant fitness. If these two hubs were to modify the effects of perturbations solely through baseline fitness, we might expect many of the same gRNAs to interact with both hubs. Instead, most gRNAs interact with only one of the two hubs, implying that a factor other than baseline segregant fitness is involved. Supporting this point, distinct CellMap enrichments are detected for the Chr X and XIII hubs.

Figure 4. How are gRNAs ordered on the x axis?

The gRNAs are ordered based on the magnitude of their mean effect, with the most detrimental guides on the left. This is purely for visualization purposes, and is now noted in the figure legend.

P. 34. 83 control gRNAs are mentioned but not described in the text. What are these?

The control gRNAs mentioned in the Methods are the same as the control gRNAs referenced in the second paragraph of the Results section, on page 6. These gRNAs target intergenic and noncoding

regions, and should have no effect on fitness even when they are induced. Text has been added to page 36 to more clearly explain this.

P. 34-35. The normalization procedure could be simplified. The authors essentially measure relative fitness with respect to the 83 control gRNAs, except the fitness of these control gRNAs now takes some non-zero (but arbitrary) value. It would be more natural and simpler to set the fitness of these control gRNAs to zero in both CON and each of ATC conditions.

The average fitness of the control gRNAs is already approximately zero in all conditions using the current normalization procedure. Also, we appreciate the point that Reviewer #1 is making, but maintain we are measuring fitness with respect to the overall population, not relative to specific control gRNAs or control lineages. For these reasons, we did not implement the suggested change.

We want to express our gratitude to Reviewer #1 for their thoughtful comments.

Reviewer #2 (Remarks to the Author):

I would like to thank the authors for their very detailed response, which in my opinion has helped improve the manuscript and clarify several points. This will in turn help readers better understand the strengths and limitations of this study. I can't help feeling that the initial submission could have been more carefully checked for errors, as the change in Figure 4 very well exemplifies. I would also like to thank the authors for sharing their analysis code and having added the bare minimum amount of guidance for anyone wishing to review it and perhaps reuse it (although I could see that virtually all paths are still absolute and therefore not going to work on someone else's system).

Thank you for this positive feedback. We appreciate Reviewer #2's help in improving this study.

REVIEWERS' COMMENTS

Reviewer #1 (Remarks to the Author):

The authors have addressed all of our comments.